# HOTA: HAMILTONIAN FRAMEWORK FOR OPTIMAL TRANSPORT ADVECTION

**Nazar Buzun**
AXXX,* Innopolis University [†]
n.buzun@seevia.ai

**Daniil Shlenskii**
AXXX*, Applied AI Institute[‡]
d.shlenskii@seevia.ai

**Maxim Bobrin**
AXXX*, Computational Imaging Lab[§]
m.bobrin@seevia.ai

**Dmitry V. Dylov**
Applied AI Institute[‡],
Computational Imaging Lab[§], AXXX*
d.dylov@seevia.ai

## ABSTRACT

Optimal transport (OT) has become a natural framework for guiding the probability flows. Yet, the majority of recent generative models assume trivial geometry (e.g., Euclidean) and rely on strong density-estimation assumptions, yielding trajectories that do not respect the true principles of optimality in the underlying manifold. We present Hamiltonian Optimal Transport Advection (HOTA), a Hamilton–Jacobi–Bellman based method that tackles the dual dynamical OT problem explicitly through Kantorovich potentials, enabling efficient and scalable trajectory optimization. Our approach effectively evades the need for explicit density modeling, performing even when the cost functionals are non-smooth. Empirically, HOTA outperforms all baselines in standard benchmarks, as well as in custom datasets with non-differentiable costs, both in terms of feasibility and optimality.
Code: https://github.com/nazarblch/HOTA
Project page: https://seevia.ai/hota

## 1 INTRODUCTION

*Static (Monge-Kantorovich)* optimal transport was originally considered as the main framework for comparing and finding a cost-minimizing coupling between distributions (Villani, 2008), while optimality was mainly measured through the boundary marginals. Development of efficient and scalable OT solvers (Cuturi, 2013; Peyré et al., 2019) popularized OT across different areas, such as generative modeling (Makkuva et al., 2020; Korotin et al., 2022; Buzun et al., 2024; Choi & Choi, 2024), computational biology (Bunne et al., 2022), graphics (Bonneel & Digne, 2023), high-energy physics (Nathan T. Suri, 2024), and reinforcement learning (Klink et al., 2022; Asadulaev et al., 2024; Bobrin et al., 2024; Rupf et al., 2025). However, one crucial limitation of static formulation is its inability to produce non-straight paths, which completely ignores the underlying geometry of the manifold of the data. In classical OT, the underlying geometric structure is solely determined by the choice of cost function (*e.g.*, , Euclidean distance), inherently limiting the capacity for fine-grained control over the trajectories. We refer to (Montesuma et al., 2024; Pereira & Amini, 2025) for recent overview of practical applications of OT and to Villani (2008); Santambrogio (2016); Peyré et al. (2019) for a formal treatment.

On the other hand, the *dynamical* OT paradigm, developed by Benamou & Brenier (2000), recasts static OT as a time variational problem on the space of probability paths, effectively incorporating time variable and enabling more nuanced control over optimal trajectories (*e.g.*, through velocity, acceleration, length, or energy over the paths). Importantly, such formulation enables one to directly

---

*AXXX, Moscow, Russia

[†]Research Center of the Artificial Intelligence Institute, Innopolis University, Innopolis, Russia

[‡]Applied AI Institute, Moscow, Russia

[§]Computational Imaging Lab, Moscow, Russia

operate on manifolds of non-trivial geometry, whenever the underlying space contains curvature, obstacles, or is defined through potentials. This formulation is closely connected to stochastic optimal control (SOC), where trajectories are stochastic yet must still maintain optimality, a problem class known as the generalized Schrödinger bridge (GSB) Liu et al. (2022); Bartosh et al. (2024).

The GSB problem generalizes the classical Schrödinger bridge by incorporating an arbitrary potential function $U(x)$ into its cost functional. This potential allows the model to incorporate rich, application-specific physics and constraints, steering optimal paths away from high-cost regions and towards low-cost valleys. The ability to handle *non-differentiable potentials* significantly expands its applicability, as many real-world constraints and penalties are inherently discontinuous or non-smooth. In computational biology, simulating molecular pathways with hard-core repulsion and steric clashes is represented by non-smooth potentials (Bunne et al., 2022). While the paper uses a static OT formulation, GSB approach could model the entire most probable path a cell takes from health to a drug-affected state. Another relevant scenario arises when the potential $U(x, t)$ is defined by a large, non-differentiable system, such as a black-box Large Language Model (LLM) or a complex neural network where propagating gradients through the entire system is computationally prohibitive or impossible Krishnamoorthy et al. (2023).

A common strategy for GSB involves solving the dual formulation via Hamilton-Jacobi-Bellman (HJB) equations, which provide a flexible and a theoretically grounded framework for deriving optimal trajectories (Liu et al. (2022), Neklyudov et al. (2024)). These methods parameterize the cost through a Lagrangian, enforcing optimality via the minimization of kinetic energy or using other path-based penalties. The dual approach is motivated by the intractability of directly minimizing the primal problem, which includes integral cost with, probably, non-smooth potentials $U(x)$. Such transformation yield a tractable objective that depends only on the value function and its derivatives and permits unbiased estimation from samples. We obtain a density-free objective that naturally incorporates non-smooth potentials and enables the use of dynamic programming for efficient multi-step trajectory optimization.

While HJB-based approaches yield theoretically sound solutions, they suffer from critical drawbacks: (1) *unstable optimization* dynamics, leading to high-variance gradients and poor sample efficiency in high dimensions, and (2) the *absence of a strict terminal distribution matching criterion*, resulting in inexact couplings. In the current work, we study the Generalized Schrödinger Bridge problem between two measures, where the underlying geometry is defined through potentials. We propose a new HJB-based framework that explicitly solves the GSB task, resolves the learning stability problems of the previous approaches, and ensures exact terminal distribution matching through a theoretically grounded objective. We conduct extensive empirical evaluations on existing low-dimensional physically inspired benchmarks, as well as in the high-dimensional generative setting, including image-to-image translation. In short, our contributions are as follows:

- Dual reformulation of GSB that binds Kantorovich potentials with an HJB value function, turning the HJB constraint into a built-in regularizer. This yields a stable objective amenable to gradient-based optimization in challenging non-Euclidean and non-smooth settings;
- We design a specialized training procedure with data replay and target network stabilization, enabling the effective joint learning of the Kantorovich potential and the optimal drift policy;
- HOTA attains state-of-the-art empirical results in a diverse set of tasks, demonstrating both better feasibility (exact marginal matching) and optimality (cost along trajectories) compared to current GSB solvers.

## 2 RELATED WORK

**Diffusion Models and Matching Algorithms.** Diffusion models have emerged as powerful tools for generative modeling by prescribing the time evolution of marginal distributions. Matching algorithms, such as Action Matching (Neklyudov et al. (2024)) and Flow Matching (Lipman et al. (2023)), learn stochastic differential equations (SDEs) that align with prescribed probability paths (Blessing et al., 2025). These methods typically assume explicit or implicit intermediate densities of the flow, whereas our approach (HOTA) optimizes a complete stochastic path from source to target distributions.

**Generalized Schrödinger Bridge.** The GSB problem extends the classical Schrödinger Bridge (SB) by incorporating a state cost function, which penalizes or rewards specific trajectories. Prior methods

for solving GSB, such as DeepGSB (Liu et al., 2022) and NLSB Koshizuka & Sato (2022), suffer from critical limitations. Particularly, absence of a strict terminal matching criterion leads to inexact couplings between the target distributions. Furthermore, they rely on explicit density estimation during their iterative procedure, which introduces additional complexity and instability, especially in high-dimensional spaces. A recent approach GSBM (Liu et al., 2024) follows an alternating optimization scheme: in the first stage, it learns the drift field $v_t$ while keeping the marginal distributions $\rho_t(x_t)$ fixed, using a Flow Matching-style objective. In the second stage, it updates the marginals conditioned on the boundary-coupled distribution $\rho_t(x_t \mid x_0, x_1)$, which is defined via the previously learned drift. While GSBM demonstrates strong empirical performance, it imposes two limitations: 1) it requires the state cost function $U(x_t)$ to be differentiable everywhere, and 2) it learns the conditional marginals $\rho(x_t \mid x_0, x_1)$ with suboptimal distributions on pairs $(x_0, x_1)$, which may lead to a biased $\rho(x_t)$ model. Moreover, as our experiments show, this alternating optimization is computationally expensive, requiring multiple iterative updates between the drift and marginals to converge.

**Stochastic Optimal Control.** The connection between GSB and stochastic optimal control (SOC) has been explored in prior works (Theodorou et al. (2010); Levine (2018)). SOC formulations often relax hard distributional constraints into soft terminal costs, which can introduce bias or require adversarial training (Liu et al. (2022)). Recently introduced Adjoint Matching approach (Domingo-Enrich et al., 2024a) and Stochastic Optimal Control matching (SOCM) (Domingo-Enrich et al., 2024b) address several existing limitations, but still produce highly unstable variance estimations. Our method provides a natural way to preserve the feasibility via Kantorovich potential sum.

## 3 FROM THE GSB PROBLEM TO THE DUAL OBJECTIVE

The core of our approach begins with the *primal* formulation of the Generalized Schrödinger Bridge (GSB) problem. We consider a controlled Markov process $x_t$ and seek the stochastic control policy $v_t$ that minimizes the expected cost of transporting a source distribution $\alpha$ to a target distribution $\beta$:

$$\inf_{v_t} \mathbb{E}\left[\int_0^1 \mathcal{L}(t, x_t, v_t)\, dt\right], \quad \mathcal{L}(t, x_t, v_t) = \frac{\|v_t\|^2}{2} + U(x_t), \quad x_0 \sim \alpha,\ x_1 \sim \beta. \tag{1}$$

The potential term $U(x_t)$ usually characterizes the geometry of the space. But in addition, we can also include some physical properties of the flow, *e.g.*, entropic penalty or "mean-field" interaction (Liu et al., 2022). Thus, the optimal trajectories are curved to avoid regions with high values of $U(x_t)$. The conditional map $\mu(\cdot|x)$ induced by the optimal drift $v_t^*$ that minimizes the primal objective (1) is, in general, stochastic. This means that for a given starting point $x \sim \alpha$, the endpoint $x_1$ is a random variable distributed according to $\mu(\cdot|x)$. This formulation, where the transport plan is described by a kernel $\mu(\cdot|x)$ rather than a deterministic function, directly relates our problem to the framework of *weak OT* (Gozlan et al., 2014). To handle the terminal constraint $x_1 \sim \beta$, we relax it by a Lagrange multiplier, which takes the form of a Kantorovich potential function $g(y)$ (ref. Definition 2.4 from Gozlan et al. (2014)). This relaxation yields the following saddle-point problem:

$$\inf_{v_t} \sup_{g \in L_1(\beta)} \left\{ \mathbb{E}\left[\int_0^1 \mathcal{L}(t, x_t, v_t)\, dt - g(x_1)\right] + \mathbb{E}_{y \sim \beta}[g(y)] \right\}. \tag{2}$$

The marginality requirement of the distribution of $x_1$ is ensured by the potential difference $\mathbb{E}_\beta g(y)$ and $\mathbb{E}_\alpha \mathbb{E}_{y \sim \mu(x)} g(y)$, which tends to infinity otherwise. Under suitable regularity conditions, we can interchange the infimum and supremum. The inner minimization over controls $v_t$ for a fixed potential $g$ defines the stochastic $c$-transform of $g$:

$$g^c(x) = \inf_{v_t} \mathbb{E}\left[\int_0^1 \mathcal{L}(t, x_t, v_t)\, dt - g(x_1)\ \middle|\ x_0 = x\right]. \tag{3}$$

The function $g^c(x)$ can be interpreted as the optimal cost-to-go from the initial state $x$ when the terminal cost is given by $-g(x_1)$. This yields the dual problem, which is a maximization over the Kantorovich potential function $g$:

$$\sup_{g \in L_1(\beta)} \left\{ \mathbb{E}_{x \sim \alpha}[g^c(x)] + \mathbb{E}_{y \sim \beta}[g(y)] \right\}. \tag{4}$$

This dual formulation is central to our method. Neural networks can effectively solve high-dimensional OT problems by learning the Kantorovich potentials, which maximize the dual objective

(Korotin et al. (2022), Buzun et al. (2024)). But unlike classical OT, we need to minimize the cost throughout the full trajectory $x_t$, $t \in [0,1]$ with the objective (3). In order to circumvent this challenge, we will address the dynamic programming principles and define the value function. Let for any $0 \le t \le 1$ and $x \in \mathbb{R}^d$, the value function satisfies:

$$s(t, x) = \inf_{v_\tau} \mathbb{E}\left[\int_t^1 \mathcal{L}(\tau, x_\tau, v_\tau)\mathrm{d}\tau - g(x_1) \,\Big|\, x_t = x\right], \tag{5}$$

such that $g^c(x) = s(0, x)$ and the boundary condition at time point $t = 1$ is

$$\forall x \in \mathbb{R}^d : s(1, x) = -g(x). \tag{6}$$

To derive a computationally tractable objective, we must now specify the stochastic dynamics governing the state evolution $x_t$. We focus on the widely used formulation of a diffusion process:

$$\mathrm{d}x_t = v(t, x_t)\mathrm{d}t + \sigma(t, x_t)\mathrm{d}W_t, \tag{7}$$

where $v(t, x_t)$ is the controllable drift and $\sigma(t, x_t)$ is the diffusion coefficient. In Appendix F, we provide the complete derivation for a more general class of dynamics, demonstrating the broader applicability of our framework. When the dynamics are given by the diffusion process (7), the function $s(t, x)$ satisfies the Hamilton-Jacobi-Bellman (HJB) equation (Fleeting & Soner, 2006). The solution to this equation subsequently enables the evaluation of the infimum in problem (3). By definition, the HJB equation is

$$-\partial_t s(t, x) = -\frac{1}{2}\|\nabla_x s(t, x)\|^2 + U(x) + \frac{\sigma^2}{2}\mathrm{tr}\{\nabla^2 s(t, x)\}. \tag{8}$$

Representation of the Lagrange function as a sum of kinetic and potential energy (1) allows us to find the minimum by control $(v_t)$ in explicit form, such that $v_t^* = -\nabla_x s(t, x_t)$. Combining the potential optimization objective (4) with the characterization of the $c$-transform via the HJB equation and its boundary condition (6), we arrive at the dual formulation of the GSB problem. This result is formally stated in the following theorem:

**Theorem 1** (Dual GSB problem). *Given distributions $\alpha, \beta \in \mathcal{P}(\mathbb{R}^d)$ and stochastic dynamics (7) with $v(t, x) = -\nabla_x s(t, x)$, the GSB problem (1) admits the following formulation:*

$$\max_{s(1,\cdot) \in L_1(\beta)} \left\{ \mathbb{E}_{x_0 \sim \alpha}\left[\int_0^1 \mathcal{L}(t, x_t, -\nabla_x s)\, dt + s(1, x_1)\right] - \mathbb{E}_{y \sim \beta}\, s(1, y)\right\}, \tag{9}$$

*where $s(t, x) \in C^{1,2}([0, 1] \times \mathbb{R}^d)$ satisfies the HJB PDE (8) $\forall t \in [0, 1]$ and $\forall x \in \mathbb{R}^d$.*

The first expression (9) plays the role of a discriminator and guarantees matching the target distribution $\beta$, and the second one (8) is responsible for the optimality of trajectories. A detailed proof is provided in Appendix E. Additionally, Appendix G contains a convergence analysis, quantifying the deviation of the learned function $s(t, x)$ from the viscosity solution in terms of the HJB residual and the potential matching loss.

By means of the optimized function $s(t, x)$ we can generate the OT trajectories using the Euler-Maruyama algorithm:

$$x_{t+\Delta t} = -\nabla_x s(t, x_t)\Delta t + \sigma \Delta W, \quad x_0 \sim \alpha. \tag{10}$$

Unlike most other methods, here we do not need to model the intermediate density of the $x_t$ ($t \in (0, 1)$) distribution, which greatly simplifies the learning process, but we need to store the generation history in a replay buffer for more stable HJB optimization in high-dimensional spaces.

## 4 METHOD

To find a stable and balanced solution $s(t, x)$ for the given GSB problem (1), we can follow a composite approach that combines optimal control (via HJB PDE constraints) and RL techniques (policy-based trajectory optimization). We approximate the value function using a parametric model $s_\theta(t, x)$. We have to maximize the potential matching functional (9) subject to the constraint that $s_\theta(t, x)$ satisfies the HJB PDE. For that divide the time interval $[0, 1]$ into $T$ time steps and simulate

$n$ trajectories $\{t_0^k, x_0^k, \ldots, t_T^k, x_T^k\}_{k=1}^n$ using initial $\alpha$ distribution and Euler-Maruyama method (10). Sample also $n$ points $y_k$ from the target distribution $\beta$ and compute the potential matching loss as

$$L_{\text{pot}}(s_\theta) = \frac{1}{n} \sum_{k=1}^n s_\theta(1, x_T^k) - \frac{1}{n} \sum_{k=1}^n s_\theta(1, y^k). \tag{11}$$

The dual objective in Theorem 1 decomposes into the integral cost and the terminal potential difference. Under the HJB constraint enforced by $L_{\text{hjb}}$, the cost term is already accounted for and minimized for the given terminal condition. Consequently, maximizing $L_{\text{pot}}$ under the HJB constraint directly optimizes the full dual objective, allowing us to omit the explicit integral term from the potential matching loss.

The HJB PDE must hold for all $t \in [0, 1]$ and $x \in \mathbb{R}^d$, but in practice, for more effective training, the training data should be sampled in the region of the flow (trajectories) concentration (according to Liu et al. (2022)). We enforce this by linear interpolation between datasets from $\alpha$ and $\beta$ as a rough estimation of the flow region and subsequently use the replay buffer $\mathcal{B}$ to collect points from the previously obtained trajectories. Using data samples $\{t^k, x^k\}_{k=1}^n$ from $\mathcal{B}$ or the linear interpolation we compute HJB residual loss as

$$L_{\text{hjb}}(s_\theta, \overline{s}) = \frac{1}{n} \sum_{k=1}^n \left( \frac{\partial s_\theta^k}{\partial t} - \frac{1}{2} \|\nabla_x \overline{s}^k\|^2 + U(x^k) + \frac{\sigma^2}{2} \text{tr}\{\nabla^2 \overline{s}^k\} + \lambda_a \|a^k\| \right)^2 \tag{12}$$

$$+ \frac{1}{n} \sum_{k=1}^n \left( \frac{\partial \overline{s}^k}{\partial t} - \frac{1}{2} \|\nabla_x s_\theta^k\|^2 + U(x^k) + \frac{\sigma^2}{2} \text{tr}\{\nabla^2 s_\theta^k\} + \lambda_a \|a^k\| \right)^2, \tag{13}$$

where $s_\theta^k = s_\theta(t^k, x^k)$ and $\overline{s}^k = \overline{s}(t^k, x^k)$ denotes the target model. The target model $\overline{s}$ is updated as an Exponential Moving Average (EMA) of the main model parameters: $\overline{\theta} \leftarrow \gamma \overline{\theta} + (1 - \gamma) \theta$. And $a^k$ is angular acceleration defined as

$$a^k = \frac{d}{dt} \frac{\nabla s_\theta(t^k, x^k)}{\|\nabla s_\theta(t^k, x^k)\|}. \tag{14}$$

---

**Algorithm 1** HOTA: Hamiltonian framework for Optimal Transport Advection

---

1: **Input**: model $s_\theta$, optimizer s_opt, distributions $\alpha$ and $\beta$, potential $U(x)$, diffusion coef. $\sigma$.
2: **Hyperparameters**: train steps $N$, iterpolation sample steps $N_0$, temporal discretization $T$, batch size $n$, hjb-loss weight $\lambda_{\text{hjb}}$, acceleration coef. $\lambda_a$, learning rate lr, gradients scale EMA coef. $\tau$.
3: **Initialize** target model $\overline{s}$; replay buffer $\mathcal{B} \leftarrow \emptyset$; gradients scale $\zeta \leftarrow 1.0$
4: **for** iteration $i = 1$ to $N$ **do**
5:     Sample train data $\{x_0^k\}_{k=1}^n \sim \alpha$; $\{y^k\}_{k=1}^n \sim \beta$
6:     **if** $i < N_0$ **then**
7:         Sample times $\{t^k\}_{k=1}^n \sim U(0, 1)$
8:         For $1 \le k \le n$ set $x^k = x_0^k \cdot (1 - t^k) + y^k \cdot t^k$
9:     **else**
10:         Sample $\{t^k, x^k\}_{k=1}^n \sim \mathcal{B}$
11:     **end if**
12:     Generate $n$ trajectories $\{t_0^k, x_0^k, \ldots, t_T^k, x_T^k\}_{k=1}^n$ using current policy $v_t = -\nabla s(t, x)$
13:     Add one of the resulting trajectories $\{t_0^k, x_0^k, \ldots, t_T^k, x_T^k\}$ to $\mathcal{B}$, where $1 \le k \le n$
14:     **Compute gradients:**
15:     $g_{\text{hjb}} = \nabla_\theta L_{\text{hjb}}(s_\theta, \overline{s}, \{t^k, x^k\}_{k=1}^n)$
16:     $g_{\text{pot}} = \nabla_\theta L_{\text{pot}}(s_\theta, \{x_T^k\}_{k=1}^n, \{y^k\}_{k=1}^n)$
17:     **Update Parameters:**
18:     Compute norms $G_{\text{hjb}} = \|g_{\text{hjb}}\|_2$ and $G_{\text{pot}} = \|g_{\text{pot}}\|_2$
19:     EMA update of gradients scale $\zeta = \tau G_{\text{pot}}/G_{\text{hjb}} + (1 - \tau)\zeta$
20:     Sum the gradients $g = g_{\text{pot}} + \lambda_{\text{hjb}} \zeta g_{\text{hjb}}$
21:     Update model parameters $\theta$ with s_opt($g$)
22:     EMA update of target model $\overline{s}$
23: **end for**

---

The angular acceleration with coefficient $\lambda_a$ forces the straightening of the trajectories (optionally). We divide the model into $s_\theta$ and $\overline{s}$ as it is usually done in Reinforcement Learning (Mnih et al., 2015) to make the optimization problem more similar to regression.

As a result, our model is trained on two criteria ($L_{\text{pot}}$ and $L_{\text{hjb}}$) simultaneously, and to balance both impacts, we scale the gradients of the hjb-loss and sum them with the pot-loss:

$$\nabla_\theta L_{\text{pot}}(s_\theta) + \lambda_{\text{hjb}} \text{EMA}\left(\frac{\|\nabla_\theta L_{\text{pot}}(s_\theta)\|}{\|\nabla_\theta L_{\text{hjb}}(s_\theta, \overline{s})\|}\right) \nabla_\theta L_{\text{hjb}}(s_\theta, \overline{s}). \tag{15}$$

The complete method is implemented as shown in Algorithm 1. It effectively combines the theoretical guarantees of optimal transport with the flexibility of neural network approximations, while maintaining numerical stability through careful gradient management. The adaptive balancing of the potential matching and HJB residual losses ensures stable convergence to a solution that satisfies both the optimality conditions and the boundary constraints.

## 5 EXPERIMENTS

In this section, we evaluate our method on a series of distribution matching tasks with non-trivial geometries. In Section 5.1, we compare HOTA with state-of-the-art baselines, demonstrating its superior performance on standard benchmarks including datasets with almost non-differentiable potentials. In Sections 5.2, 5.3 we demonstrate the scalability of our approach by showcasing its effectiveness in high-dimensional settings. Finally, in Section 5.5, we ablate key components of our method.

**Evaluation.** We assess performance using two main metrics: *feasibility* and *optimality*. Feasibility reflects how well the method matches the target distribution, evaluated via Wasserstein distance with squared Euclidean cost ($W_2(T_{\#}\alpha, \beta)$), where the transport mapping $T$ uses optimized value function $s_\theta$ and simulates $x_1$ by procedure (10). Optimality measures the quality of the resulting mapping, estimated through the integral trajectory cost (1) where $x_t$ follows (7) and $v_t^* = -\nabla_x s(t, x_t)$.

**Baselines.** We use source code from GSBM repository for running it in our experiments. Other results were taken from the original papers NLOT Pooladian et al. (2024), NLSB Koshizuka & Sato (2022), DeepGSB Liu et al. (2022) where datasets were previously introduced.

**Runtime.** Our method achieves efficient high-dimensional OT computation through JAX's automatic differentiation, with minimal overhead (<5%) for computing second-order derivatives. HOTA demonstrates near-linear scalability with both data dimensionality and simulation steps, significantly outperforming GSBM with speedups of 50-100×. While this section focuses on comparing the effectiveness of these methods, we provide an analysis of their computational efficiency in Appendix B.

**Reproducibility.** To support reproducibility, we attach our full code in the supplementary material. All synthetic datasets can be generated with the provided scripts, and high-dimensional experiments use standard public benchmarks. All experiments are conducted on a single GPU A100 80GB. Complete experimental details, including hyperparameters and network architectures, are documented in Appendix A.

### 5.1 COMPARATIVE EVALUATION ON LOW-DIMENSIONAL DATA

In this section, we compare our method to previous state-of-the-art approaches on the standard benchmarks including datasets that feature almost non-differentiable potential functions. Visualizations of the datasets are provided in Figure 1.

The first three datasets—Stunnel, Vneck, and GMM—are adopted from Liu et al. (2024). These benchmarks incorporate state cost functions $U(x_t)$ that encourage the optimal solution to respect complex geometric constraints. Each dataset is designed to highlight specific capabilities of the evaluated algorithms. *Stunnel* assesses whether a method can capture drift fields that undergo rapid and localized changes. *Vneck* evaluates the ability to model drift that compresses and expands the support of marginal distributions. *GMM* tests whether the method can disambiguate closely situated points and assign them to distinct trajectories. The remaining datasets—BabyMaze, Slit, and Box (Pooladian et al. (2024))—are constructed using similar underlying principles but pose additional

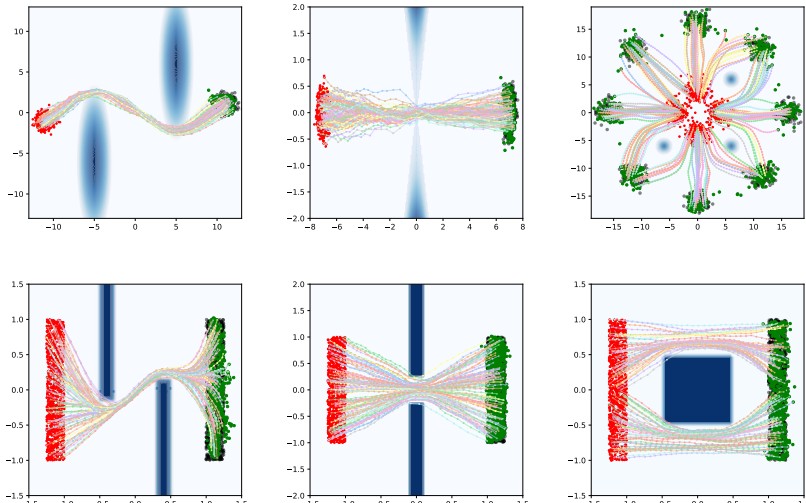

Figure 1: Evaluation of HOTA method on smooth (top) and non-smooth datasets (bottom): Stunnel, Vneck, GMM, BabyMaze, Slit, Box. Blue regions indicate high values of potential $U(x)$. Distributions $\alpha$ (red), $\beta$ (black) and the mapped $T_{\#}\alpha$ (green).

Table 1: Quantitative comparison between baseline methods and our approach, HOTA. Performance is evaluated using two criteria: *Feasibility* (how well the target distribution is covered) and *Optimality* (efficiency of the learned mapping). N/A cells indicate that original authors of particular method did not include results for those tasks. The mean and the standard deviations of our method are computed across 5 different seeds. Best values are highlighted by bold font (lower is better). Gray values correspond to the method's divergence (in terms of Feasibility).

| | Feasibility $W_2(T_{\#}(\alpha), \beta)$ | | | Optimality (integral cost) | | |
|---|---|---|---|---|---|---|
| | Stunnel | Vneck | GMM | Stunnel | Vneck | GMM |
| NLSB | 30.54 | 0.02 | 67.76 | 207.06 | 147.85 | 4202.71 |
| GSBM | 0.03 | 0.01 | 4.13 | 460.88 | 155.53 | 229.12 |
| **HOTA** | **0.006**$_{\pm 0.003}$ | **0.002**$_{\pm 0.001}$ | **0.19**$_{\pm 0.05}$ | **383.25**$_{\pm 10.5}$ | **115.09**$_{\pm 8.9}$ | **80.44**$_{\pm 2.6}$ |
| | BabyMaze | Slit | Box | BabyMaze | Slit | Box |
| NLSB | $> 1$ | 0.013 | 0.024 | N/A | N/A | N/A |
| NLOT | $> 1$ | 0.013 | 0.016 | N/A | N/A | N/A |
| GSBM | 0.01 | 0.01 | 0.02 | 6.5 | 4.9 | 3.8 |
| **HOTA** | **0.004**$_{\pm 0.003}$ | **0.0004**$_{\pm 0.0001}$ | **0.002**$_{\pm 0.001}$ | **4.87**$_{\pm 0.14}$ | **3.06**$_{\pm 0.09}$ | **2.84**$_{\pm 0.11}$ |

difficulties due to the presence of almost non-differentiable state cost functions. A summary of the quantitative results across all datasets is provided in Table 1. Gray values in this table means that the method with corresponding configuration significantly diverges from the target distribution. For this reason, we omit such comparisons with the others by the Optimality metric. Our method, HOTA, consistently outperforms existing approaches in terms of both feasibility and optimality. In particular, HOTA achieves a substantial performance gain on the GMM dataset, which may refer to its superior capability in trajectory separation for closely situated points.

In addition to the synthetic 2D benchmarks, we evaluate HOTA on a more complex problem: the transportation of 3D LiDAR point clouds (Appendix D). This setting features an irregular potential, $U(x)$, derived from real-world scanned surfaces. This experiment tests the method's robustness and ability to handle complex, state-dependent costs in an applied scenario.

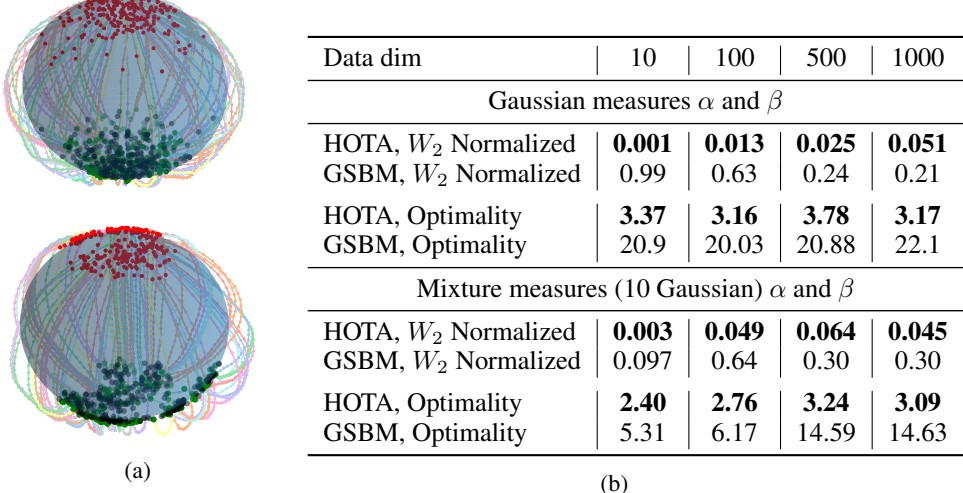

|  |  |  |  |  |
| --- | --- | --- | --- | --- |
| Data dim | 10 | 100 | 500 | 1000 |
| Gaussian measures $\alpha$ and $\beta$ | | | | |
| HOTA, $W_2$ Normalized | **0.001** | **0.013** | **0.025** | **0.051** |
| GSBM, $W_2$ Normalized | 0.99 | 0.63 | 0.24 | 0.21 |
| HOTA, Optimality | **3.37** | **3.16** | **3.78** | **3.17** |
| GSBM, Optimality | 20.9 | 20.03 | 20.88 | 22.1 |
| Mixture measures (10 Gaussian) $\alpha$ and $\beta$ | | | | |
| HOTA, $W_2$ Normalized | **0.003** | **0.049** | **0.064** | **0.045** |
| GSBM, $W_2$ Normalized | 0.097 | 0.64 | 0.30 | 0.30 |
| HOTA, Optimality | **2.40** | **2.76** | **3.24** | **3.09** |
| GSBM, Optimality | 5.31 | 6.17 | 14.59 | 14.63 |

(a)

(b)

Figure 2: (a) Visualization of our *Sphere* dataset for $N = 3$ with Gaussian (top) and Mixtute of 10 Gaussian (bottom) measures $\alpha$ and $\beta$. (b) Performance comparison between our HOTA model and a recent state-of-the-art GSBM on high-dimensional Sphere datasets. Models are evaluated on *Feasibility* (W2 Normalized) and *Optimality*. The best value for each metric is bolded.

## 5.2 SCALABILITY TO HIGH-DIMENSIONAL SPACES

In this section, we test the scalability of our method, demonstrating its stable performance in higher-dimensional settings. For this purpose, we use *Sphere* datasets parameterized by data dimensionality $N$. The source and target measures are Gaussian or a mixture of Gaussian distributions projected onto the sphere's surface. To enforce geometric constraints, we introduce a potential $U(x)$ that penalizes deviations from the manifold: $U_{\text{sphere}}(x) = C_d \mathbf{I}(\|x\| < 1)$, where $C_d > 0$ is a dimension-dependent constant. The performance of our method across varying data dimensions is shown in Figure 2 with three-dimensional case visualization. Since the $W_2$ metric becomes less informative in high dimensions, we report the normalized $W_2$. This is computed as the $W_2(T_\#(\alpha), \beta)$ distance divided by $W_2(\alpha, \beta)$ and substituting distance between two independent samples from the target distribution $(\beta, \beta')$, i.e.,

$$\text{Normalized}(W_2) = \frac{W_2(T_\#(\alpha), \beta) - W_2(\beta, \beta')}{W_2(\alpha, \beta)}. \tag{16}$$

This normalization provides a better measure of convergence relative to the intrinsic spread of the target data. Notably, HOTA demonstrates robust and stable performance as the dimensionality $N$ increases.

## 5.3 HIGH-DIMENSIONAL OPINION DEPOLARIZATION

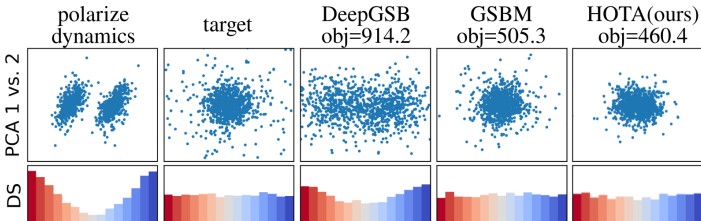

Figure 3: Opinion prediction in dimension 1000 and correspondent directional similarities (DS). Polarize dynamics is the opinion distribution formed without control. The target measure $\beta$ is multivariate Gaussian with mean 0 and standard deviation $4I$.

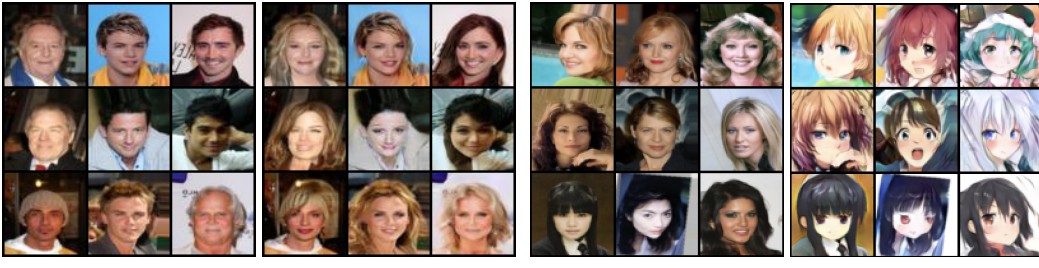

(a) Male (CelebA) → Female (CelebA).  (b) Female (CelebA) → Anime

Figure 4: Examples of learned transport maps on (left) CelebA male→female and (right) CelebA female→Anime Faces. HOTA shifts domain-specific attributes (style/texture/color) while maintaining content such as identity cues, expression, and pose.

Our model employs the polarization mechanism from Liu et al. (2022), which builds on the framework of the political parties introduced by Gaitonde J. (2021). During each iteration $t$, every agent observes a shared random stimulus $\xi_t \in \mathbb{R}^{1000}$ drawn independently from the current opinion distribution $p_t$. The agents then update their positions using a dynamic response mechanism. This mechanism drives an agent's movement based on its alignment with other agents regarding a piece of information. If an agent's evaluation of the information aligns with another agent's, the interaction attracts them toward that agent's direction. Conversely, if their evaluations differ, the interaction causes repulsion. This formulation captures the psychological tendency for individuals to gravitate toward aligned viewpoints while resisting opposing perspectives, ultimately leading to group polarization. In this case, we consider the stochastic process with polarization drift ($f_{\text{polarize}}$, defined in Liu et al. (2022)):

$$dx_t = v(t, x_t)dt + f_{\text{polarize}}(x_t; p_t, \xi_t)dt + \sigma dW. \tag{17}$$

The control task is to depolarize the final distribution $x_1$ and match with the target measure $\beta = \mathcal{N}(0, 4I)$. The optimization *objective* here is the integral kinetic energy (1), where $\mathcal{L} = v_t^2/2$ and $U(x_t) = 0$. We analyze the distribution of angular separations between opinion vectors (directional similarities Schweighofer S. (2020)). A more uniform angular distribution indicates reduced polarization, while peaked distributions suggest stronger factional divisions. The comparison in terms of the task objective and directional similarities is presented in Figure 3.

## 5.4 IMAGE-TO-IMAGE TRANSLATION

To validate HOTA's capability on high-dimensional, real-world data, we evaluate it on image-to-image translation tasks. Specifically, we learn transport maps (1) between male and female faces within the CelebA dataset (Liu et al. (2015)) and (2) between CelebA (female faces) and the Anime[1] dataset, translating from a photographic to an illustrative domain. These tasks require learning a transport map that modifies core semantic attributes (e.g., style, texture, and coloration) across domains while preserving the underlying content and structure of each input image.

Table 2: FID scores for image-to-image experiments.

| Task | Model | FID |
|---|---|---|
| Male (CelebA)→Female (CelebA) (64×64) | CycleGAN Zhu et al. (2017) | 12.94 |
| | NOT Korotin et al. (2022) | 11.96 |
| | EUOT Choi & Choi (2024) | 8.44 |
| | HOTA (ours) | **6.28** |
| Female (CelebA)→Anime (64×64) | CycleGAN Zhu et al. (2017) | 20.80 |
| | NOT Korotin et al. (2022) | 18.28 |
| | ENOT Buzun et al. (2024) | 13.12 |
| | HOTA (ours) | **11.67** |

---

[1]kaggle.com/datasets/reitanaka/alignedanimefaces

Qualitative results in Figure 4 show that the transport learned by HOTA produces samples that closely match the target distribution while preserving key semantic attributes of the input. Quantitatively, we measure performance using the Fréchet Inception Distance (FID), which assesses the quality and diversity of translated images relative to the target distribution. As reported in Table 2, HOTA achieves competitive FID scores, outperforming strong baselines, including recent Neural Optimal Transport methods (NOT Korotin et al. (2022), ENOT Buzun et al. (2024)).

## 5.5 ABLATION STUDY

Table 3 presents comparison of the full HOTA model against variants without the replay buffer $\mathcal{B}$ that stores simulation history, target EMA model $\bar{s}(t, x)$, or the adaptive gradient balancing by means of $\alpha$ (15), evaluating as previously feasibility and optimality metrics across Stunnel, Vneck, and GMM datasets. The full HOTA achieves strong metric scores, while removing the buffer severely degrades feasibility in Vneck and GMM and increases costs in Stunnel. Disabling gradient balancing harms feasibility in Stunnel and GMM. The results highlight the buffer's critical role in maintaining feasibility and the nuanced trade-offs between gradient balancing and transport efficiency across different scenarios.

Table 3: Comparison of HOTA method against variants without the replay buffer $\mathcal{B}$, adaptive gradient balancing and target EMA model. Best values are highlighted by bold font (lower is better). Gray values correspond to the method's divergence (in terms of Feasibility).

|  | Feasibility $W_2(T_\#(\alpha), \beta)$ | | | Optimality (integral cost) | | |
|---|---|---|---|---|---|---|
|  | Stunnel | Vneck | GMM | Stunnel | Vneck | GMM |
| HOTA | **0.006** | **0.002** | **0.19** | **383.25** | 115.09 | **80.44** |
| HOTA w/o buffer | 0.076 | 16.47 | 1.248 | 706.89 | 82.49 | 121.6 |
| HOTA w/o grad. balancing | 3.60 | 0.026 | 2.64 | 385.62 | **109.25** | 72.77 |
| HOTA w/o EMA model | 0.018 | 0.004 | 0.65 | 408.60 | 109.88 | 97.30 |

## 6 LIMITATIONS AND FUTURE WORK

While HOTA exhibits strong performance, we observed sensitivity to certain network design choices—particularly the Fourier feature encoding of time, a commonly used technique in models that estimate ODE drifts. Additionally, because the value function in our framework must simultaneously support optimal control estimation and serve as a Kantorovich potential, it requires a network architecture capable of aggregating rich temporal and spatial information. Despite these limitations, HOTA opens new possibilities for solving optimal transport problems in real-world applications, such as computational biology, robotics, and generative modeling. Future work will focus on extending the theoretical guarantees to weaker regularity assumptions and on incorporating more structured neural architectures to further improve scalability and stability.

## 7 CONCLUSION

In this work, we introduced HOTA, a new OT method based on the Hamilton–Jacobi–Bellman (HJB) framework for solving the Generalized Schrödinger Bridge problem. We demonstrated that HOTA consistently outperforms recent state-of-the-art methods on standard benchmarks and scales effectively to high-dimensional settings. Remarkably, it works for non-smooth potentials and with non-differentiable cost functions, yielding robust performance gain in terms of strictly defined concepts of feasibility and optimality.

## ACKNOWLEDGMENT

The work was supported by the Ministry of Economic Development of the Russian Federation (agreement No. 139-10-2025-034 dd. 19.06.2025, IGK 000000C313925P4D0002).

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

## A  ADDITIONAL EXPERIMENTAL DETAILS

**Hyperparameters.**   Table 4 summarizes the hyperparameters used for each dataset presented in the paper. Note that the Sphere datasets, which are parameterized by data dimensionality, share all hyperparameters except for the potential weight, which may take the value 10 for low dimensions and 50 for high ones. In our image-to-image experiments, we replace the MLP backbone with a ResNet architecture and adopt a specialized configuration of Fourier feature embeddings tailored for images.

Table 4: Hyperparameters used for each dataset presented in the paper.

| Hyperparameter | Stunnel | Vneck | GMM | BabyMaze | Slit | Box | Sphere | Opinion | LiDAR | Image2Image |
|---|---|---|---|---|---|---|---|---|---|---|
| MLP hidden layers | | | | $[512, 512, 512, 1]$ | | | | | | − |
| Fourier frequencies | | | | $\{1, \dots, 20\}$ | | | | | | $\{10^{-4}, \dots, 1.0\}$ |
| optimizer | | | | Adam with cosine annealing ($\alpha = 1 \times 10^{-2}$) | | | | | | |
| initial learning rate | | | | $5 \times 10^{-4}$ | | | | | | |
| Adam $[\beta_1, \beta_2]$ | | | | $[0.9, 0.99]$ | | | | | | $[0.5, 0.9]$ |
| # training iterations | | | | 70000 | | | | | | 100000 |
| batch size | | | | 1024 | | | | | | 64 |
| scale EMA, $\tau$ | | | | 0.9 | | | | | | |
| model EMA, $\gamma$ | | | | 0.99 | | | | | | |
| # control steps | | | | 30 | | | | | | |
| diffusion coef., $\sigma$ | 0.3 | 0.2 | 0.1 | 0.03 | 0.05 | 0.03 | 0.01 | 0.5 | 0.2 | 0.05 |
| control weight, $\lambda_{\text{hjb}}$ | 1.0 | 2.0 | 0.7 | 0.5 | 2.0 | 0.3 | 0.4 | 0.4 | 1.0 | 0.5 |
| acc. weight, $\lambda_a$ | 0.0001 | 0.001 | 0.2 | 0.05 | 0.001 | 0.01 | 0 | 0 | 0.001 | 0 |
| potential weight | 25 | 1000 | 25 | 10 | 30 | 700 | $\{10, 50\}$ | 20 | 0.1 | 0 |

**Network.**   In all our experiments (except image datasets), we employ a simple MLP, augmented with Fourier feature encoding of the time component. For general time embeddings of the form $\text{emb}(t) = \sin(f \cdot t + \varphi)$, the time derivative is given by $\partial_t \text{emb}(t) = f \cdot \cos(f \cdot t + \varphi)$. As the frequency $f$ increases, the magnitude of this derivative also grows, potentially leading to numerical instability—especially when the time derivative of the network is explicitly involved in the objective. This issue has been previously discussed in Lu & Song (2024). To address this, we restrict the frequency range to $[1, 20]$ and normalize the resulting Fourier features by dividing by the corresponding frequencies.

**Impact of acceleration and control weights.**   We have evaluated the influence of the balancing coefficient $\lambda_{\text{hjb}}$ and the acceleration term $\lambda_a |a|$ in the loss $L_{\text{hjb}}$. The latter penalizes changes in angular velocity to straighten trajectories—our results show that increasing $\lambda_a$ improves trajectory optimality while introducing a slight bias in matching the target distribution $\beta$, as reflected in the feasibility metric. In the GMM task, due to the specificity of the dataset and the divergence of trajectories in different directions, a small penalization of acceleration also improves feasibility. Simultaneously, we optimize the losses $L_{\text{pot}}$ and $L_{\text{hjb}}$, scaling the HJB-loss gradients for stability before summing them with the POT-loss gradients, and we investigate the sensitivity of learning to $\lambda_{\text{hjb}}$ and its impact on performance in the Stunnel and GMM tasks (Figure 5).

**Image-to-image translation.**   For image-to-image translation tasks we parameterize the value function $s_\theta(t, x)$ using a 5-layer ResNet architecture from WGAN-QC (Liu et al. (2019)). The network uses a channel progression of $[64, 128, 256, 512, 512]$, with skip connections to facilitate gradient flow. A critical design choice is the integration of the temporal dimension: the time embedding with Fourier features is concatenated to the input of each residual block. We optimize the parameters using the Adam with momentum $(\beta_1, \beta_2) = (0.5, 0.9)$. For more stable training, we also used $L_1$ loss in HJB objective 12, scaled the kinetic energy term $v_t^2/2$ by a constant 0.01 and ran simulations in both forward (source to target) and backward (target to source) directions to populate the replay buffer more effectively. The other training hyperparameters are listed in Table 4.

## B  SCALABILITY AND RUNTIME

While parametrizing the value function $s(t, x)$ with a neural network requires computing its second-order derivatives, our method incurs minimal overhead in practice. This efficiency stems from two factors: the overhead scales linearly with the number of simulation steps, and JAX's automatic

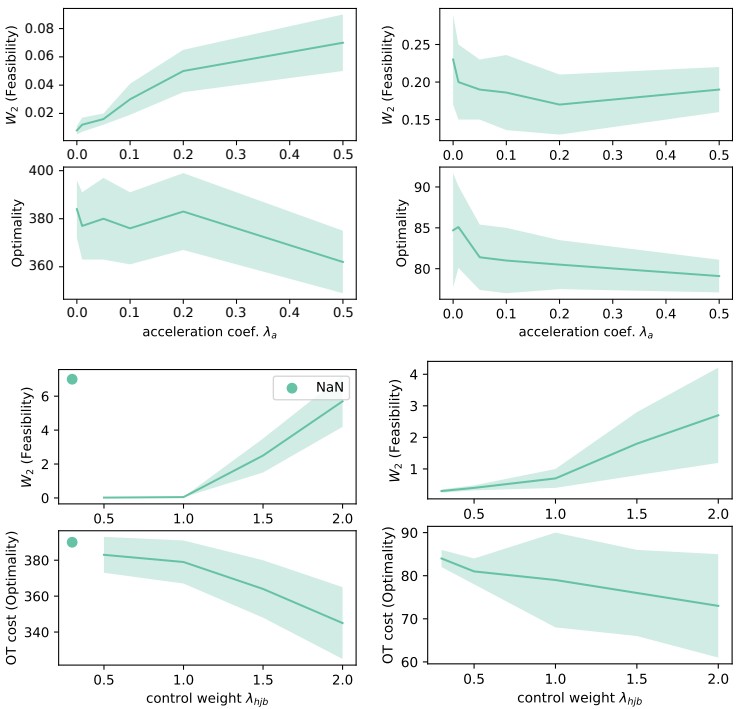

Figure 5: Impact of acceleration coefficient $\lambda_a$ and control weight $\lambda_{\text{hjb}}$. Left: Stunnel, right: GMM datasets. The results show a tradeoff: increasing $\lambda_a$ or $\lambda_{\text{hjb}}$ improves optimal transport (OT) cost but reduces feasibility. Larger instability arises for both excessively low and high values of $\lambda_{\text{hjb}}$.

differentiation computes $\nabla s$ and $\Delta s$ in near-identical time to $s(t, x)$ itself (consistently below 5% overhead). This was verified using a 4-layer MLP across dimensions from $10^2$ to $10^5$. The primary speedup is achieved through JAX's scan-based optimization of the simulation loops.

To further illustrate efficiency, we conducted experiments on our Sphere dataset. We fixed the number of training iterations at 70,000 and varied the data dimensionality (with simulation steps fixed at 30) as well as varied the number of simulation steps (with data dimensionality fixed at 1000). The results in the Tables 5 below show that HOTA scales well (near-linear) with both increasing dimensionality and an increasing number of simulation steps. HOTA achieves significant speedups over GSBM due to (1) a simpler optimization objective and (2) better dimensional scalability. Speedup factors are calculated as the ratio of GSBM's runtime to HOTA's runtime. Notably, GSBM also encounters out-of-memory (OOM) errors beyond 5,000 dimensions under 24GB GPU memory constraints, while HOTA maintains stable performance through 9,000 dimensions.

Table 5: Runtime dependence on dimension and simulation steps.

| Data dim | 1k | 3k | 5k | 7k | 9k |
|---|---|---|---|---|---|
| HOTA time | 4.4 min | 7.1 min | 9.9 min | 12.8 min | 15.7 min |
| GSBM Time | 241 min | 620 min | 1233 min | OOM | OOM |
| Speedup ($\times$) | 54.8 | 87.3 | 124.5 | – | – |

| Simulation steps | 10 | 20 | 30 | 40 | 50 |
|---|---|---|---|---|---|
| HOTA time | | 3.4 min | 3.7 min | 4.4 min | 5.8 min | 7.1 min |

| Datasets | 2D benchmark | 1000D opinion depolarization |
|---|---|---|
| GSBM time | 30-45 min | 1495 min |
| HOTA (ours) time | 6-7 min | 26 min |

## C  HOTA OBJECTIVES CONVERGENCE

In this section we provide an additional empirical analysis of our HOTA convergence. We tracked the evolution of the losses $L_{\text{hjb}}$ and $L_{\text{pot}}$ during training on the Stunnel and Slit datasets. The results in Figure 6 show that both losses consistently decrease towards zero, meaning that the learned value function $s_\theta$ increasingly satisfies the conditions of Theorem 1. In particular, $L_{\text{pot}}$ corresponds to $\mathbb{E}_\alpha s(1, x_1) - \mathbb{E}_\beta s(1, y)$ in equation 9, so its convergence indicates that the terminal potential $s_\theta(1, \cdot)$ approximately optimizes the dual GSB problem and enforces marginal matching. At the same time, $L_{\text{hjb}}$ is the squared residual of the HJB PDE in Theorem 1, and its decay shows that $s_\theta(t, x)$ becomes an increasingly accurate solution of the associated optimal control problem. Taken together, these trends empirically support that HOTA is optimizing the dual formulation of Theorem 1 while approximately satisfying the HJB constraint along the learned trajectories, rather than only fitting terminal marginals.

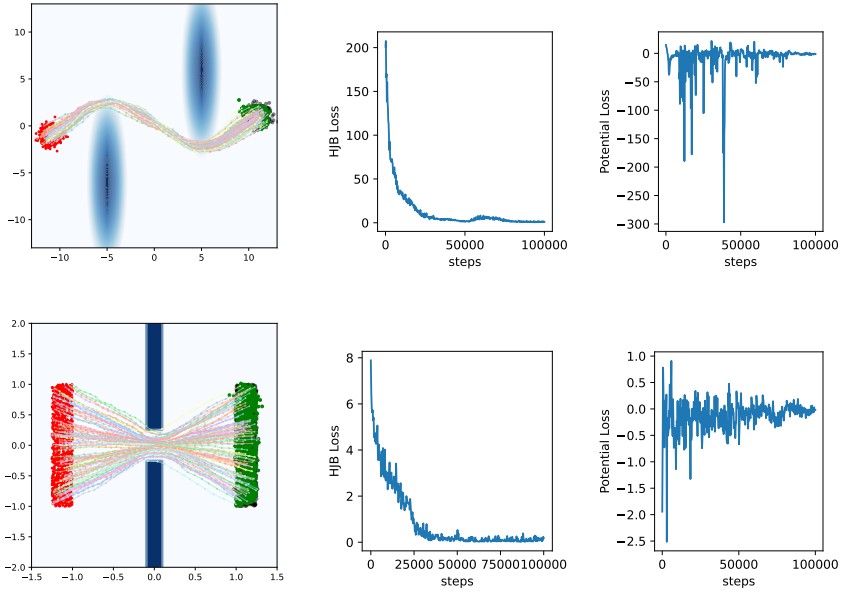

Figure 6: Empirical convergence justification of $L_{\text{hjb}}$ and $L_{\text{pot}}$ losses. Top row: Stunnel (smooth $U(x)$), bottom row: Slit (non-smooth $U(x)$) datasets.

## D  EXPERIMENTS WITH LiDAR DATASET

In this section we provide an additional evaluation of HOTA on surfaces observed through LiDAR scans from (Liu et al., 2024) (see Figure 7). We adopt the state cost used in (Liu et al., 2024):

$$U(x) = L_{\text{manifold}}(x) + L_{\text{height}}(x), \ L_{\text{manifold}}(x) = \|\pi(x) - x\|_2^2, \ L_{\text{height}}(x) = \exp\big(\pi^{(z)}(x)\big), \quad (18)$$

where $\pi(x)$ projects $x$ onto an approximate tangent plane fitted by $k$-nearest neighbors, and $\pi^{(z)}(x)$ refers to the $z$-coordinate of $\pi(x)$, $i.e.$, its height.

Table 7 (left) shows that HOTA achieves the lowest integral trajectory cost, improving upon GSBM and DeepGSB, which indicates more efficient transport on this complex real-world manifold. At the same time, HOTA attains a small Wasserstein distance, demonstrating accurate matching of the target distribution, as qualitatively illustrated in Figure 7. Together, these results confirm that HOTA extends effectively beyond synthetic benchmarks to irregular LiDAR-derived geometries.

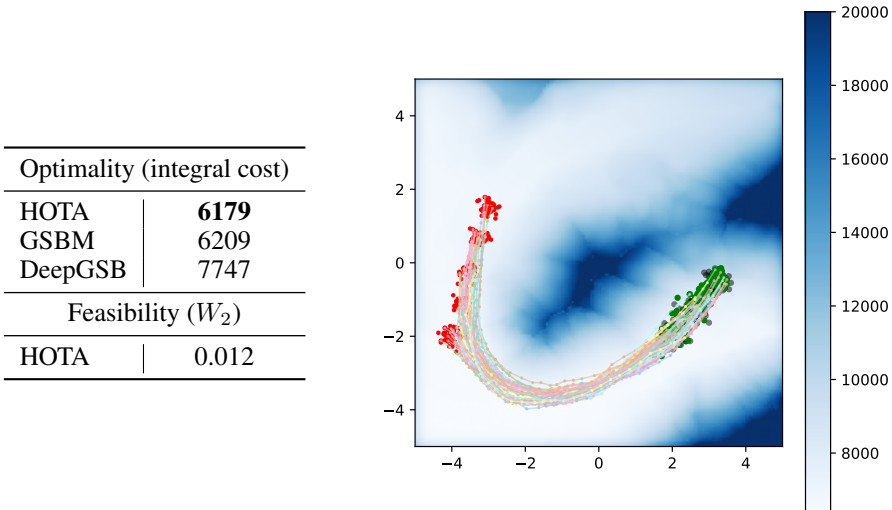

| Optimality (integral cost) | |
|---|---|
| HOTA | **6179** |
| GSBM | 6209 |
| DeepGSB | 7747 |
| Feasibility ($W_2$) | |
| HOTA | 0.012 |

Figure 7: LiDAR dataset: quantitative results (left) and qualitative visualization of HOTA trajectories on the scanned surface (right).

## E   PROOF OF THEOREM 1 (DUAL FORMULATION OF GSB)

We prove, in the **first step** , the equivalence between the GSB (stochastic control formulation) and its dual formulation using Kantorovich-style duality. Remind that we consider the stochastic process $x_t$ (7) with conditions $x_0 \sim \alpha$, $x_1 \sim \beta$, a control function $v(t, x_t)$, and Brownian motion $\sigma(t, x_t)dW_t$. The **primary problem** of GSB optimization is:

$$\inf_v \mathbb{E}\left[\int_0^1 \mathcal{L}(t, x_t, v_t)dt\right] \quad \text{s.t.} \quad x_0 \sim \alpha, \ x_1 \sim \beta, \tag{19}$$

where in the particular case $\mathcal{L}(t, x, v) = v^2/2 + U(x)$. Since the stochastic process $x_t$ starts from $x_0 \sim \alpha$, the primal problem is equivalent to:

$$\inf_v \left(\mathbb{E}\left[\int_0^1 \mathcal{L}(t, x_t, v_t)dt\right] + \sup_{g \in L_1(\beta)} \left(-\mathbb{E}[g(x_1)] + \mathbb{E}_\beta[g(y)]\right)\right), \tag{20}$$

where the supremum over $g$ enforces the constraint $x_1 \sim \beta$ (via Lagrange duality). Rewrite the Lagrangian problem as

$$\inf_v \sup_{g \in L_1(\beta)} \left(\mathbb{E}\left[\int_0^1 \mathcal{L}(t, x_t, v_t)dt - g(x_1)\right] + \mathbb{E}_\beta[g(y)]\right). \tag{21}$$

Assuming strong duality holds under mild regularity conditions (e.g., $\mathcal{L}$ convex in $v$, $\alpha, \beta$ absolutely continuous, see Villani (2008)), we swap inf and sup:

$$\sup_{g \in L_1(\beta)} \left(\inf_v \mathbb{E}\left[\int_0^1 \mathcal{L}(t, x_t, v_t)dt - g(x_1)\right] + \mathbb{E}_\beta[g(y)]\right). \tag{22}$$

Moreover, since the optimal control can be chosen independently for each starting point $x$, we have

$$\mathbb{E}_{x \sim \alpha}\left[\inf_v \mathbb{E}\left[\int_0^1 \mathcal{L}\, dt - g(x_1) \,\Big|\, x_0 = x\right]\right] = \inf_v \mathbb{E}\left[\int_0^1 \mathcal{L}\, dt - g(x_1)\right],$$

where the infimum on the right is taken over all Markovian controls. This identity follows from the dynamic programming principle and the fact that the value function depends only on the current state. By the definition of $c$-conjugate transform (3):

$$\mathbb{E}_{x \sim \alpha} \inf_v \mathbb{E}\left[\int_0^1 \mathcal{L}(t, x_t, v_t)dt - g(x_1) \,\Big|\, x_0 = x\right] = \mathbb{E}_{x \sim \alpha} g^c(x). \tag{23}$$

Thus, the **dual problem** becomes:

$$\sup_{g \in L_1(\beta)} \left( \mathbb{E}_\alpha[g^c(x)] + \mathbb{E}_\beta[g(y)] \right). \tag{24}$$

In the **second step**, find the optimal control solution $v^*(t, x)$ by means of the dynamic programming principle. Define the value function $s(t, x)$ that, for any $0 \le t \le \tau \le 1$, satisfies:

$$s(t, x) = \inf_v \mathbb{E} \left[ \int_t^\tau \mathcal{L}(z, x_z, v_z) dz + s(\tau, x_\tau) \,\Big|\, x_t = x \right]. \tag{25}$$

Applying Ito's formula to $s(\tau, x_\tau)$, we obtain that

$$ds(\tau, x_\tau) = \partial_\tau s d\tau + \nabla s \cdot dx_\tau + \frac{1}{2} \text{tr}(\sigma^2 \nabla^2 s) d\tau \tag{26}$$

$$= \left( \partial_\tau s + \nabla s^T v_\tau + \frac{1}{2} \text{tr}(\sigma^2 \nabla^2 s) \right) d\tau + \nabla s^T \sigma dW_\tau. \tag{27}$$

Consider the evolution of the value between times $t$ and $\tau$:

$$s(\tau, x_\tau) - s(t, x_t) = \int_t^\tau \left( \partial_z s + \nabla s \cdot v_z + \frac{1}{2} \text{tr}(\sigma^2 \nabla^2 s) \right) dz + \int_t^\tau \nabla s^T \sigma dW. \tag{28}$$

Based on the martingale property of Ito integrals ($\mathbb{E}[\int \nabla s \cdot \sigma dW | x_t = x] = 0$), it holds that

$$\mathbb{E}[s(\tau, x_\tau) | x_t = x] = s(t, x) + \mathbb{E} \left[ \int_t^\tau \left( \partial_z s + \nabla s \cdot v_z + \frac{1}{2} \text{tr}(\sigma^2 \nabla^2 s) \right) dz \right]. \tag{29}$$

Substitute back into dynamic programming and plug the last expression into the equation (25):

$$s(t, x) = \inf_v \mathbb{E} \left[ \int_t^\tau \mathcal{L}(z, x_z, v_z) dz + s(t, x) + \int_t^\tau \left( \partial_z s + \nabla s^T v_\tau + \frac{1}{2} \text{tr}(\sigma^2 \nabla^2 s) \right) dz \right]. \tag{30}$$

Cancel $s(t, x)$ from both sides and divide by $(\tau - t)$:

$$0 = \inf_{v(s,t)} \frac{1}{\tau - t} \mathbb{E} \left[ \int_t^\tau \left( \mathcal{L}(z, x_z, v_z) + \partial_z s + \nabla s^T v_z + \frac{1}{2} \text{tr}(\sigma^2 \nabla^2 s) \right) dz \right]. \tag{31}$$

Take the limit $\tau \downarrow t$ to derive the HJB equation for a general Lagrangian $\mathcal{L}$

$$0 = \inf_v \left\{ \mathcal{L}(t, x, v) + \partial_t s + \nabla s^T v + \frac{1}{2} \text{tr}(\sigma^2 \nabla^2 s) \right\}. \tag{32}$$

Identify the optimal control for the particular $\mathcal{L}(t, x, v) = v^2/2 + U(x)$. The infimum is attained when $v^* = -\nabla s$, yielding the final result of Theorem 1.

For the HJB equation to have a *unique solution* (in the viscosity sense), we require *coercivity* (Theorem 4.1 (Fleeting & Soner, 2006)) of the Hamiltonian for some constants $C_1 > 0$ and $C_2 \ge 0$

$$H(x, \nabla s, \nabla^2 s) = \frac{1}{2} \|\nabla_x s\|^2 - U(x) - \frac{\sigma^2}{2} \text{tr}\{\nabla^2 s\} \tag{33}$$

$$\ge C_1(\|\nabla s\|) - C_2(1 + \|x\| + \|\nabla^2 s\|) \tag{34}$$

The term $\|\nabla_x s\|^2$ dominates for large values, so in case $U(x)$ is bounded and $\sigma > 0$, the solution is unique.

## F    GENERAL HJB

To derive the solution for $g^c(x)$, one may consider the expectation over general stochastic processes (not necessarily diffusion or one driven by an SDE). In a general case, we define the value function as:

$$s(t, x) = \inf_v \mathbb{E} \left[ \int_t^1 L(\tau, x_\tau, v_\tau) d\tau - g(x_1) \,\Big|\, x_t = x \right], \tag{35}$$

with a terminal condition $s(1, x) = -g(x)$. Then, $g^c(x) = s(0, x)$. Using the Dynamic Programming Principle, one can obtain the HJB equation

$$-\partial_t s(t, x) = \inf_v \{\mathbb{E}L(t, x, v) + \mathcal{A}^v s(t, x)\}, \tag{36}$$

where $\mathcal{A}^v$ is the infinitesimal generator of the process:

$$\mathcal{A}^v s(t, x) = \lim_{h \to 0} \frac{\mathbb{E}[s(t + h, x_{t+h}) - s(t, x) \mid x_t = x, v_t = v]}{h}, \tag{37}$$

and the optimal control satisfies:

$$v^* = \operatorname{argmin}_v \{\mathbb{E}L(t, x, v) + \mathcal{A}^v s(t, x)\}. \tag{38}$$

At this step we need to decide on the type of the random process $x_t$. For diffusion (in the main text, we often assume that $dx_t = v_t\, dt + \sigma\, dW_t$):

$$\mathcal{A}^v s = v \cdot \nabla_x s + \frac{1}{2} \operatorname{tr}(\sigma\sigma^T \nabla_x^2 s). \tag{39}$$

If the control $v_t$ is given by a general function $v_t = f(x_t, \varepsilon_t)$, with $\varepsilon_t$ representing external randomness, e.g., noise, the HJB equation becomes:

$$-\partial_t s(t, x) = \inf_f \{\mathbb{E}_{\varepsilon_t}[L(t, x, f(x, \varepsilon_t))] + \mathbb{E}_{\varepsilon_t}[f(x_t, \varepsilon_t)] \cdot \nabla_x s\}. \tag{40}$$

Note that the diffusion-based solution is a special case motivated by the tractability in optimizing the value function. The SDE structure is particularly advantageous in numerical implementations, as it allows for efficient gradient-based optimization and aligns with existing literature on diffusion-based generative models.

## G THEORETICAL CONVERGENCE ANALYSIS

We aim to prove that, under suitable assumptions, the error between the neural network approximation $s_\theta$ and the true viscosity solution $s^*$ of the Hamilton-Jacobi-Bellman (HJB) equation is controlled by the PDE residual and the terminal loss.

Recall the dual formulation of GSB (Theorem 1):

$$\sup_{g \in L^1(\beta)} \{\mathbb{E}_{x \sim \alpha}[g^c(x)] + \mathbb{E}_{y \sim \beta}[g(y)]\}, \tag{41}$$

where the $c$-transform $g^c$ is defined via the value function $s(t, x)$ of the stochastic control problem:

$$g^c(x) = \inf_v \mathbb{E}\left[\int_0^1 \mathcal{L}(t, x_t, v_t)\, dt - g(x_1) \,\Big|\, x_0 = x\right]. \tag{42}$$

The HJB equation holds for $s$:

$$\partial_t s + H(t, x, \nabla s, \nabla^2 s) = 0 \quad \text{in } (0, 1) \times \Omega \tag{43}$$

where the Hamiltonian is:

$$H(t, x, p, M) = -\frac{1}{2}\|p\|^2 + U(x) + \frac{\sigma^2}{2}\operatorname{tr}(M) \tag{44}$$

Assume the following regularity conditions hold:

1. The HJB equation has a unique viscosity solution $s^* \in C^{1,2}([0, 1] \times \Omega)$ and both functions $s^*, s_\theta \in C^{1,2}([0, 1] \times \overline{\Omega})$;

2. $\Omega \subset \mathbb{R}^d$ is a bounded domain with $C^2$ boundary, and both $s^*$ and $s_\theta$ satisfy the same boundary condition (Dirichlet or Neumann) such that all boundary terms vanish in integrations by parts;

3. The diffusion coefficient satisfies $\sigma > 0$. The diffusion term here provides regularization and prevents singularities in the optimal control;

4. The HJB objective residual:

$$\mathcal{R}(s_\theta) = \partial_t s_\theta + H(t, x, \nabla s_\theta, \nabla^2 s_\theta), \text{ is uniformly bounded: } \|\mathcal{R}(s_\theta)\|_{L^\infty} \leq \epsilon_{\text{hjb}}. \quad (45)$$

The following theorem establishes the error estimate under the assumption of bounded deviation between the terminal functions $s_\theta(1, x)$ and $s^*(1, x)$. While this assumption provides a useful starting point for analysis, it is restrictive and may not fully capture the practical behavior of learned approximations. Below, we will consider a relaxation of this condition, extending the results to more realistic and weaker assumptions that align with empirical training dynamics.

**Theorem 2.** *Under Assumptions 1-4, there exists a constant $C > 0$ such that:*

$$\|s_\theta - s^*\|_{L^\infty} \leq C \left( \epsilon_{\text{hjb}} + \|s_\theta(1, \cdot) - s^*(1, \cdot)\|_{L^\infty} \right) \quad (46)$$

*Moreover, the optimal control converges as:*

$$\int_0^1 \int_\Omega \|v_\theta(t, x) - v^*(t, x)\|^2 \mathrm{d}x\mathrm{d}t \leq \left( \|s_\theta(1, \cdot) - s^*(1, \cdot)\|_{L^2}^2 + \|R_\theta\|_{L^2}^2 \right) \quad (47)$$

*where $v_\theta = -\nabla s_\theta$, $v^* = -\nabla s^*$.*

*Proof.* The true solution $s^*$ satisfies:

$$\partial_t s^* + H(t, x, \nabla s^*, \nabla^2 s^*) = 0. \quad (48)$$

Use notation

$$\phi = s_\theta - s^*. \quad (49)$$

Subtracting the first equation from the second, we get the PDE for the error $\phi$:

$$\partial_t \phi + H(t, x, \nabla s_\theta, \nabla^2 s_\theta) - H(t, x, \nabla s^*, \nabla^2 s^*) = \mathcal{R}(s_\theta). \quad (50)$$

Let's analyze the difference in Hamiltonians (44). Since $U(x)$ cancels out in the difference, we have:

$$H(t, x, \nabla s_\theta, \nabla^2 s_\theta) - H(t, x, \nabla s^*, \nabla^2 s^*) \quad (51)$$

$$= -\frac{1}{2}\|\nabla s_\theta\|^2 + \frac{1}{2}\|\nabla s^*\|^2 + \frac{\sigma^2}{2} \text{tr}(\nabla^2 s_\theta - \nabla^2 s^*) \quad (52)$$

$$= -\frac{1}{2}\langle \nabla s_\theta + \nabla s^*, \nabla \phi \rangle + \frac{\sigma^2}{2} \text{tr}(\nabla^2 \phi). \quad (53)$$

Where we used $\|\nabla s_\theta\|^2 - \|\nabla s^*\|^2 = \langle \nabla s_\theta + \nabla s^*, \nabla s_\theta - \nabla s^* \rangle = \langle \nabla s_\theta + \nabla s^*, \nabla \phi \rangle$.

Define a vector field $b(t, x)$ and a constant $A$:

$$b(t, x) = -\frac{1}{2}(\nabla s_\theta + \nabla s^*), \quad A = \frac{\sigma^2}{2}. \quad (54)$$

Then equation (50) becomes:

$$\partial_t \phi + b(t, x) \cdot \nabla \phi + A \, \text{tr}(\nabla^2 \phi) = \mathcal{R}(s_\theta). \quad (55)$$

This is a linear parabolic PDE for $\phi$. Use the Feynman-Kac formula and represent the PDE as:

$$\phi(t, x) = \mathbb{E}\left[ \phi(1, x_1) + \int_t^1 \mathcal{R}(s_\theta)(r, X_r) \, dr \,\bigg|\, X_t = x \right], \quad (56)$$

where $dX_r = b(r, X_r)dr + \sigma dW_r$. Taking absolute values and suprema:

$$|\phi(t, x)| \leq \mathbb{E}\left[ |\phi(1, x_1)| + \int_t^1 |\mathcal{R}(s_\theta)(r, X_r)| \, dr \,\bigg|\, X_t = x \right]. \quad (57)$$

Since $\|\mathcal{R}(s_\theta)\|_{L^\infty} \leq \epsilon_{\text{hjb}}$, we have

$$\|\phi\|_{L^\infty([0,1]\times\Omega)} \leq \|\phi(1, \cdot)\|_{L^\infty(\Omega)} + \epsilon_{\text{hjb}}. \quad (58)$$

This is a key inequality. It shows that the error over the entire space-time domain is bounded by the error at the terminal time plus the accumulated PDE residual.

Next, obtain the control convergence bound. The optimal controls are $v^* = -\nabla s^*$ and $v_\theta = -\nabla s_\theta$. Therefore, $v_\theta - v^* = -\nabla\phi$. We need to bound $\|\nabla\phi\|_{L_2}^2$.

Multiply the PDE (55) by $\phi$ and integrate over $\Omega$:

$$\int_\Omega \phi\partial_t\phi\,dx + A\int_\Omega \phi\Delta\phi\,dx + \int_\Omega \phi\,b\cdot\nabla\phi\,dx = \int_\Omega \phi\mathcal{R}(s_\theta)\,dx. \tag{59}$$

Using $\int \phi\partial_t\phi = \frac{1}{2}\frac{d}{dt}\|\phi\|_{L^2}^2$ and integrating by parts the diffusion term (boundary terms vanish by Assumption 2):

$$\int_\Omega \phi\Delta\phi\,dx = -\int_\Omega |\nabla\phi|^2\,dx = -\|\nabla\phi\|_{L^2}^2. \tag{60}$$

For the drift term we apply Cauchy–Schwarz and Young's inequality: for any $\varepsilon > 0$,

$$\left|\int_\Omega \phi\,b\cdot\nabla\phi\,dx\right| \le \|b\|_{L^\infty}\|\phi\|_{L^2}\|\nabla\phi\|_{L^2} \le \frac{\|b\|_{L^\infty}^2}{2\varepsilon}\|\phi\|_{L^2}^2 + \frac{\varepsilon}{2}\|\nabla\phi\|_{L^2}^2. \tag{61}$$

Similarly, the right-hand side is bounded by

$$\int_\Omega \phi\mathcal{R}(s_\theta)\,dx \le \|\phi\|_{L^2}\|\mathcal{R}(s_\theta)\|_{L^2} \le \frac{1}{2}\|\phi\|_{L^2}^2 + \frac{1}{2}\|\mathcal{R}(s_\theta)\|_{L^2}^2. \tag{62}$$

Collecting these estimates, we obtain

$$\frac{1}{2}\frac{d}{dt}\|\phi\|_{L^2}^2 + A\|\nabla\phi\|_{L^2}^2 \le \frac{\|b\|_{L^\infty}^2}{2\varepsilon}\|\phi\|_{L^2}^2 + \frac{\varepsilon}{2}\|\nabla\phi\|_{L^2}^2 + \frac{1}{2}\|\phi\|_{L^2}^2 + \frac{1}{2}\|\mathcal{R}(s_\theta)\|_{L^2}^2. \tag{63}$$

Choose $\varepsilon = A$ (so that the term $\frac{\varepsilon}{2}\|\nabla\phi\|^2$ can be absorbed into the left-hand side) and rearrange:

$$\frac{1}{2}\frac{d}{dt}\|\phi\|_{L^2}^2 + \frac{A}{2}\|\nabla\phi\|_{L^2}^2 \le C_1\|\phi\|_{L^2}^2 + \frac{1}{2}\|\mathcal{R}(s_\theta)\|_{L^2}^2, \tag{64}$$

where $C_1 = \frac{\|b\|_{L^\infty}^2}{2A} + \frac{1}{2}$. Now integrate from $t$ to $1$:

$$\frac{1}{2}\|\phi(1,\cdot)\|_{L^2}^2 - \frac{1}{2}\|\phi(t,\cdot)\|_{L^2}^2 + \frac{A}{2}\int_t^1 \|\nabla\phi\|_{L^2}^2\,ds \le C_1\int_t^1 \|\phi\|_{L^2}^2\,ds + \frac{1}{2}\int_t^1 \|\mathcal{R}(s_\theta)\|_{L^2}^2\,ds. \tag{65}$$

Dropping the nonpositive term $-\frac{1}{2}\|\phi(t)\|_{L^2}^2$ and applying Grönwall's inequality (or simply bounding $\int_t^1 \|\phi\|_{L^2}^2\,ds$ by the terminal value plus the integral of the residual, using the representation from the Feynman–Kac formula), we finally obtain

$$\int_0^1 \|\nabla\phi\|_{L^2}^2\,dt \le C\left(\|\phi(1,\cdot)\|_{L^2}^2 + \int_0^1 \|\mathcal{R}(s_\theta)\|_{L^2}^2\,dt\right), \tag{66}$$

for some constant $C$ depending on $A$, $\|b\|_{L^\infty}$, and the length of the interval. This yields the desired control convergence bound (equation 47).

$\square$

In the previous proposition we showed that, once a candidate network $s_\theta$ approximately solves the HJB equation, the space–time error between $s_\theta$ and the true solution $s^*$ can be controlled in terms of the terminal discrepancy and the HJB residual. Concretely, we obtained an estimate of the form

$$\|s_\theta - s^*\|_{L^\infty([0,1]\times\Omega)} \lesssim \|\phi(1,\cdot)\|_{L^\infty(\Omega)} + \|\mathcal{R}(s_\theta)\|_{L^\infty}, \qquad \phi := s_\theta - s^*. \tag{67}$$

However, from a learning perspective, we never observe $s^*$ directly, and we do not minimize $\|\phi(1,\cdot)\|$ in training. Instead, we control two quantities: the dual objective gap (Kantorovich-type functional between $s_\theta$ and $s^*$), and the PDE residual loss: $\|\mathcal{R}(s_\theta)\|_{L^2}$ over $[0,1]\times\Omega$.

To turn these training losses into rigorous guarantees on the learned dynamics and controls, we must therefore express the terminal error

$$\|\phi(1,\cdot)\|_{L^2(\Omega)}^2 = \|s_\theta(1,\cdot) - s^*(1,\cdot)\|_{L^2(\Omega)}^2 \tag{68}$$

directly in terms of the objective gap. Denote $g(x) = s(1, \cdot)$ and write the (unregularized) Kantorovich dual

$$J(g) = \mathbb{E}_{x \sim \alpha}[g^c(x)] + \mathbb{E}_{y \sim \beta}[g(y)], \tag{69}$$

where $g^c$ is the $c$-transform (equivalently, the HJB value at $t = 0$). Let assume the comparison principle holds for the associated HJB so that $g^c(x) = s^g(0, x)$. For the OT dual $J$ is concave but typically not strongly concave and $g^*$ is only unique up to an additive constant. A common way is to regularize $g$:

$$J_\lambda(g) := \mathbb{E}_\alpha[g^c(x)] + \mathbb{E}_\beta[g(y)] - \frac{\lambda}{2}\|g\|_{\mathcal{H}}^2, \qquad \lambda > 0, \tag{70}$$

where $(\mathcal{H}, \langle \cdot, \cdot \rangle_{\mathcal{H}})$ is a Hilbert space (e.g. $\|\nabla g\|_{L^2}$). Then $J_\lambda$ is $\lambda$-strongly concave in $\mathcal{H}$. For all $g, h \in \mathcal{H}$

$$J_\lambda(h) \leq J_\lambda(g) + \langle DJ_\lambda(g), h - g \rangle_{\mathcal{H}} - \frac{\lambda}{2}\|h - g\|_{\mathcal{H}}^2. \tag{71}$$

We next provide a trajectory-based estimate for the quality of a learned terminal potential. Given an approximate HJB solution $s_\theta$ with a uniformly bounded residual, we relate the regularized dual objective value $J_\lambda$ of the induced terminal potential $g_\theta = -s_\theta(1, \cdot)$ to an explicitly computable surrogate $\widetilde{J}_\lambda(\theta)$ obtained by simulating the closed-loop diffusion driven by $-\nabla s_\theta$. Under a realizability assumption for the maximizer of the regularized dual and a continuous embedding $\mathcal{H} \hookrightarrow L^2(\beta)$, this yields an $L^2(\beta)$ error bound on the terminal potential.

**Theorem 3** (Trajectory-form bound for the regularized dual potential). *Assume $s_\theta \in C^{1,2}([0,1] \times \mathbb{R}^d)$, $s_\theta(1, \cdot) \in \mathcal{H}$, and $\|\mathcal{R}(s_\theta)\|_{L^\infty((0,1) \times \mathbb{R}^d)} \leq \varepsilon_{\text{hjb}}$. Define the closed-loop diffusion*

$$dx_t^\theta = -\nabla s_\theta(t, x_t^\theta)\, dt + \sigma\, dW_t, \quad x_0^\theta \sim \alpha, \tag{72}$$

*assuming it is well-posed and the expectations below are finite. Let $\mathcal{H}$ be a Hilbert space with continuous embedding into $L^2(\beta)$, i.e. $\|h\|_{L^2(\beta)} \leq C_{\text{emb}}\|h\|_{\mathcal{H}}$. Let $g^* \in \arg\max_{g \in \mathcal{H}} J_\lambda(g)$ and define $s^*(1, \cdot) := -g^*(\cdot)$. Assume that $g^*$ is representable by the parametric model, i.e., there exists $\theta^*$ such that*

$$s_{\theta^*}(1, \cdot) = -g^*, \qquad \text{in particular } s_{\theta^*}(1, \cdot) \in \mathcal{H}. \tag{73}$$

*Define the trajectory-form surrogate*

$$\widetilde{J}_\lambda(\theta) := \mathbb{E}\left[s_\theta(1, x_1^\theta) + \int_0^1 \left(\tfrac{1}{2}\|\nabla s_\theta\|^2 + U\right)(t, x_t^\theta)\, dt\right] - \mathbb{E}_\beta[s_\theta(1, y)] - \frac{\lambda}{2}\|s_\theta(1, \cdot)\|_{\mathcal{H}}^2. \tag{74}$$

*Then*

$$\|s_\theta(1, \cdot) - s^*(1, \cdot)\|_{L^2(\beta)} \leq C_{\text{emb}}\sqrt{\frac{2}{\lambda}\left(\widetilde{J}_\lambda(\theta^*) - \widetilde{J}_\lambda(\theta) + 4\varepsilon_{\text{hjb}}\right)}. \tag{75}$$

*Proof.* Fix $\theta$ and set

$$g_\theta := -s_\theta(1, \cdot) \in \mathcal{H}. \tag{76}$$

Let $s^{g_\theta}$ denote the (exact) solution of the terminal-value HJB problem

$$\mathcal{R}(s) = 0 \quad \text{on } (0,1) \times \mathbb{R}^d, \qquad s(1, \cdot) = -g_\theta(\cdot), \tag{77}$$

where $\mathcal{R}$ is as in the statement,

$$\mathcal{R}(s) = \partial_t s - \tfrac{1}{2}\|\nabla s\|^2 + U + \tfrac{\sigma^2}{2}\text{tr}(\nabla^2 s). \tag{78}$$

By the definition of the $c$-transform through the HJB/value-function representation associated with equation 77, we have

$$g_\theta^c(x) = s^{g_\theta}(0, x), \qquad x \in \mathbb{R}^d. \tag{79}$$

Define, for $(t, x) \in [0, 1] \times \mathbb{R}^d$,

$$s_\theta^+(t, x) := s_\theta(t, x) + \varepsilon_{\text{hjb}}(1 - t), \qquad s_\theta^-(t, x) := s_\theta(t, x) - \varepsilon_{\text{hjb}}(1 - t). \tag{80}$$

Since $\nabla s_\theta^\pm = \nabla s_\theta$ and $\nabla^2 s_\theta^\pm = \nabla^2 s_\theta$, while $\partial_t s_\theta^+ = \partial_t s_\theta - \varepsilon_{\text{hjb}}$ and $\partial_t s_\theta^- = \partial_t s_\theta + \varepsilon_{\text{hjb}}$, it follows from equation 78 that

$$\mathcal{R}(s_\theta^+) = \mathcal{R}(s_\theta) - \varepsilon_{\text{hjb}}, \qquad \mathcal{R}(s_\theta^-) = \mathcal{R}(s_\theta) + \varepsilon_{\text{hjb}}. \tag{81}$$

Using the assumption $\|\mathcal{R}(s_\theta)\|_{L^\infty((0,1)\times\mathbb{R}^d)} \leq \varepsilon_{\text{hjb}}$, we obtain

$$\mathcal{R}(s_\theta^+) \leq 0, \qquad \mathcal{R}(s_\theta^-) \geq 0 \quad \text{on } (0,1) \times \mathbb{R}^d. \tag{82}$$

Moreover, the terminal conditions coincide:

$$s_\theta^+(1,\cdot) = s_\theta^-(1,\cdot) = s_\theta(1,\cdot) = -g_\theta(\cdot). \tag{83}$$

Assuming the comparison principle holds for equation 77 (with sub/supersolutions interpreted in the usual viscosity sense), we get

$$s_\theta^+(t,x) \leq s^{g_\theta}(t,x) \leq s_\theta^-(t,x), \qquad (t,x) \in [0,1] \times \mathbb{R}^d. \tag{84}$$

Evaluating it at $t = 0$ and using equation 80 yields

$$\|s_\theta(0,\cdot) - s^{g_\theta}(0,\cdot)\|_{L^\infty(\mathbb{R}^d)} \leq \varepsilon_{\text{hjb}}. \tag{85}$$

Combining with equation 79 gives

$$\|s_\theta(0,\cdot) - g_\theta^c(\cdot)\|_{L^\infty(\mathbb{R}^d)} \leq \varepsilon_{\text{hjb}}. \tag{86}$$

Taking expectation under $x_0^\theta \sim \alpha$ yields

$$\left| \mathbb{E}_\alpha[g_\theta^c(x_0^\theta)] - \mathbb{E}_\alpha[s_\theta(0, x_0^\theta)] \right| \leq \varepsilon_{\text{hjb}}. \tag{87}$$

Let $X^\theta$ solve the closed-loop SDE from the statement,

$$dx_t^\theta = -\nabla s_\theta(t, x_t^\theta)\, dt + \sigma\, dW_t, \qquad x_0^\theta \sim \alpha. \tag{88}$$

Applying Itô's formula to $t \mapsto s_\theta(t, x_t^\theta)$ (using $s_\theta \in C^{1,2}$) gives

$$ds_\theta(t, x_t^\theta) = \left( \partial_t s_\theta - \|\nabla s_\theta\|^2 + \tfrac{\sigma^2}{2} \operatorname{tr}(\nabla^2 s_\theta) \right)(t, x_t^\theta)\, dt + \sigma \nabla s_\theta(t, x_t^\theta) \cdot dW_t. \tag{89}$$

Using the identity obtained by rearranging equation 78,

$$\partial_t s_\theta - \|\nabla s_\theta\|^2 + \tfrac{\sigma^2}{2} \operatorname{tr}(\nabla^2 s_\theta) = \mathcal{R}(s_\theta) - \left( \tfrac{1}{2}\|\nabla s_\theta\|^2 + U \right), \tag{90}$$

integrating over $[0,1]$, taking expectation, and using $\mathbb{E}\int_0^1 \nabla s_\theta \cdot dW_t = 0$, we obtain

$$\mathbb{E}[s_\theta(0, x_0^\theta)] = \mathbb{E}\left[ s_\theta(1, x_1^\theta) + \int_0^1 \left( \tfrac{1}{2}\|\nabla s_\theta\|^2 + U \right)(t, x_t^\theta)\, dt \right] - \mathbb{E}\int_0^1 \mathcal{R}(s_\theta)(t, x_t^\theta)\, dt. \tag{91}$$

By the residual bound $\|\mathcal{R}(s_\theta)\|_{L^\infty} \leq \varepsilon_{\text{hjb}}$,

$$\left| \mathbb{E}[s_\theta(0, x_0^\theta)] - \mathbb{E}\left[ s_\theta(1, x_1^\theta) + \int_0^1 \left( \tfrac{1}{2}\|\nabla s_\theta\|^2 + U \right)(t, x_t^\theta)\, dt \right] \right| \leq \varepsilon_{\text{hjb}}. \tag{92}$$

Relate $J_\lambda(g_\theta)$ and $\widetilde{J}_\lambda(\theta)$. Combining equation 87 and equation 92 yields

$$\left| \mathbb{E}_\alpha[g_\theta^c(x_0^\theta)] - \mathbb{E}\left[ s_\theta(1, x_1^\theta) + \int_0^1 \left( \tfrac{1}{2}\|\nabla s_\theta\|^2 + U \right)(t, x_t^\theta)\, dt \right] \right| \leq 2\varepsilon_{\text{hjb}}. \tag{93}$$

Using $g_\theta = -s_\theta(1,\cdot)$, we have

$$\mathbb{E}_\beta[g_\theta] = -\mathbb{E}_\beta[s_\theta(1, y)], \qquad \|g_\theta\|_\mathcal{H} = \|s_\theta(1,\cdot)\|_\mathcal{H}. \tag{94}$$

Substituting the last expressions into the definition of $J_\lambda(g_\theta)$ gives

$$J_\lambda(g_\theta) \geq \widetilde{J}_\lambda(\theta) - 2\varepsilon_{\text{hjb}}. \tag{95}$$

Since $J_\lambda$ is $\lambda$-strongly concave on $\mathcal{H}$, for its maximizer $g^* \in \arg\max_{g \in \mathcal{H}} J_\lambda(g)$ we have

$$J_\lambda(g^*) - J_\lambda(g_\theta) \geq \frac{\lambda}{2}\|g_\theta - g^*\|_\mathcal{H}^2. \tag{96}$$

We have assumed $g^*$ is representable by the parametric model in the sense that there exists $\theta^*$ such that $s_{\theta^*}(1,\cdot) = -g^*$. Applying equation 95 to $\theta$ and to $\theta^*$, we obtain

$$J_\lambda(g^*) - J_\lambda(g_\theta) = J_\lambda(g_{\theta^*}) - J_\lambda(g_\theta) \leq \widetilde{J}_\lambda(\theta^*) - \widetilde{J}_\lambda(\theta) + 4\varepsilon_{\text{hjb}}. \tag{97}$$

It yields

$$\|g_\theta - g^*\|_\mathcal{H} \leq \sqrt{\frac{2}{\lambda}\left( \widetilde{J}_\lambda(\theta^*) - \widetilde{J}_\lambda(\theta) + 4\varepsilon_{\text{hjb}} \right)}. \tag{98}$$

Using the continuous embedding $\|h\|_{L^2(\beta)} \leq C_{\text{emb}}\|h\|_\mathcal{H}$ and equality $s_\theta(1,\cdot) - s^*(1,\cdot) = g^* - g_\theta$, we conclude

$$\|s_\theta(1,\cdot) - s^*(1,\cdot)\|_{L^2(\beta)} \leq C_{\text{emb}} \sqrt{\frac{2}{\lambda}\left( \widetilde{J}_\lambda(\theta^*) - \widetilde{J}_\lambda(\theta) + 4\varepsilon_{\text{hjb}} \right)}. \tag{99}$$

$\square$

