# OpenReview forum: "HOTA: Hamiltonian framework for Optimal Transport Advection"
_ICLR.cc/2026/Conference — ICLR 2026 Poster_

### Official Review · Reviewer_qy11 · 2025-10-29

**Soundness:** 2
**Presentation:** 2
**Contribution:** 2
**Rating:** 4
**Confidence:** 2

**Summary:**

This paper presents a Hamiltonian Framework for Optimal Transport Advection. The contribution of this work appears to be the loss functions (Eqs. 14-17) and the algorithm, which learns dynamical OT problems with a dual Hamiltonian formulation. The authors claim that the proposed approach surpasses SOTA performance in low-dimensional settings.

**Strengths:**

* The proposed method is mostly sound, with a clear presentation of the mathematical derivation.
* The outlined experiments validate the effectiveness of the formulation.

**Weaknesses:**

* The motivation, reasoning, and verification for considering the dual GSB problem are not clear and need enhancement.
* The theoretical arguments in Section 3 are somewhat well known from stochastic mechanics. Therefore, I think contributions of this work seems to be focused in the methodology part.
* Sample inefficiency. I believe Eq. (17) contains the gradient of Hessians, which cannot be easily computed in high dimensionality. The computation of Eqs. (14-15) is seemingly not sample-efficient for large models. The authors are encouraged to discuss this limitation if the method requires more training time than the benchmarks.

**Questions:**

* Can HOTA be applied to image-to-image translation tasks?

---

> ### Author Response · Authors · 2025-11-27
>
> We thank the reviewer for their time and comments!
>
> **W1: The motivation, reasoning, and verification for considering the dual GSB problem**
>
> We thank the reviewer for this remark. In response, we have clarified the motivation for the dual formulation in the revised manuscript. Specifically, we now emphasize that the dual approach is motivated by the intractability of directly minimizing the primal problem, which involves an integral cost with non-smooth potentials U(x). The dual version yield a tractable objective that depends only on the value function and its derivatives and permits unbiased estimation from samples. We obtain a density-free objective that naturally incorporates non-smooth potentials and enables the use of dynamic programming for efficient multi-step trajectory optimization.
>
> **W2: The theoretical arguments in Section 3 are somewhat well known from stochastic mechanics.**
>
> While we respectfully acknowledge the reviewer's opinion, we would like to disagree that the theoretical part is based solely on well-known results and would like to clarify our key theoretical and methodological contributions.
>
> First, unlike the classical setup in stochastic mechanics, which often involves optimizing a minimization problem with a given terminal condition, we formulate and solve the **min-max problem**.
>
> Second, we do not merely apply the well-known Hamilton-Jacobi-Bellman (HJB) equations; rather, we **link the Kantorovich dual problem** to the Generalized Schrödinger Bridge (GSB), which is a non-trivial generalization. It enables the simultaneous optimization of trajectories and guarantees matching of the target measure using only a single parametric model \(s_\theta\) for the potential.
>
> Third, as the reviewer correctly noted, a significant methodological contribution is the **specialized optimization procedure**,  which is critical for training stability in high dimensions.
>
> Finally, complementing the theoretical part, we add a  new section (Appendix H) with a **convergence analysis for our model**. This formally justifies its performance and, to the best of our knowledge, constitutes a new result for this class of problems.
>
> **W3: Sample inefficiency. I believe Eq. (17) contains the gradient of Hessians, which cannot be easily computed in high dimensionality.**
>
> We thank the reviewer for this important remark regarding computational efficiency. We agree that the computation of second derivatives (Hessians) in equations (14,15, 17) can raise concerns about scalability. However, our empirical evaluation demonstrates that the proposed method is not only practically feasible but also **significantly outperforms state-of-the-art baselines in computation speed**.
>
> As detailed in Section 5 ("Experiments") and  Appendix B ("Scalability and runtime"), our HOTA method demonstrates near-linear scalability with respect to data dimensionality and the number of simulation steps. Specifically, Table 4 presents a comparison of runtimes: HOTA achieves a **speedup of 50-100x** compared to the GSBM method (Liu et al., 2024) on high-dimensional tasks (1000-5000 dimensions). Furthermore, while GSBM encounters out-of-memory (OOM) errors beyond 5000 dimensions, HOTA maintains stable performance up to 9000 dimensions under the same GPU memory constraints.
>
> This high efficiency is achieved due to two key factors:
> 1.  The use of automatic differentiation in JAX allows for computing $\nabla s$ and $\Delta s$ in nearly the same time as the function $s(t,x)$ itself, with minimal overhead (<5%), as noted in Section 5.
> 2.  Our optimization objective is simpler and more direct compared to the iterative schemes used in other methods like GSBM, leading to faster convergence and less overall computation.
>
> **Q1: Can HOTA be applied to image-to-image translation tasks?**
>
> Yes, HOTA can be applied to image-to-image translation tasks, and we have included into the paper new experiments with images. In Appendix E, HOTA was evaluated on several domain adaptation tasks using the CelebA and Anime face datasets.
> The results demonstrate that HOTA successfully learns transport maps that transform core attributes between domains while preserving the underlying structure and similarity of the input images.
> The performance was quantitatively measured using the FID metric. As reported in Table 5, HOTA achieved good results, outperforming other baselines like CycleGAN, Neural OT methods (NOT, ENOT).
>
> However, it is important to note that generating high-quality, high-resolution images requires separate and extensive research into the selection of neural network architectures and parameter optimization methods. The results presented in the paper demonstrate the principled effectiveness of the HOTA framework for image translation, but scaling it to state-of-the-art generative modeling benchmarks would involve significant additional engineering and architectural considerations beyond the scope of the current study.

---

### Official Review · Reviewer_Z9DH · 2025-11-02

**Soundness:** 2
**Presentation:** 2
**Contribution:** 2
**Rating:** 4
**Confidence:** 4

**Summary:**

This paper proposes a method for estimating the time-dependent Kantorovich potential $s:[0,1]×R^d \rightarrow R$ using tools from optimal transport and Hamilton–Jacobi–Bellman (HJB) theory. The terminal potential $s(1,⋅)$ is learned using the Kantorovich dual formulation, and intermediate values $s(t, \cdot)$ are obtained by solving the HJB equation backward in time. The approach is demonstrated on 2D synthetic datasets, a 3D sphere-to-sphere transport, and opinion depolarization task.

**Strengths:**

1. The paper clearly states the goal of learning the full space–time potential $s(t,x)$. This only requires a single neural network parameterizes the value function.

2. The theoretical derivation is well structured, connecting the Kantorovich duality with the HJB equation in a coherent way. Visualizations of the learned transport trajectories and potentials are helpful for intuition.

**Weaknesses:**

1. **Limited originality relative to prior work.** The overall methodology, estimating a terminal potential via the Kantorovich dual and propagating it using HJB, closely resembles prior work [1]. The main difference is the inclusion of a state-dependent running cost, but the theoretical structure remains largely unchanged. As a result, the contribution appears incremental rather than conceptually novel.

2. **Concerns about algorithmic practicality.**

- Optimizing the unregularized Kantorovich dual objective is known to be unstable and highly sensitive to initialization and regularization.

- Accurately enforcing the HJB equation corresponds to solving a nonlinear PDE, which is notoriously difficult, especially in high-dimensional spaces.

- - Even small numerical errors in estimating $s(t,x)$ may lead to inaccurate gradient estimation $\nabla s(t, x)$. For instance, if value function has little oscillatory or noisy estimates, that would change the gradient estimation of the value function $s$ drastically, leading to inaccurate estimation of the control.

- The method follows an Eulerian perspective, requiring accurate approximation of $s(t,x)$ across the entire state space. This becomes computationally infeasible beyond very low-dimensional domains, suggesting limited scalability beyond 2D or 3D, or toy datasets.

3. **Experimental validation is limited and low-dimensional.**

- All experiments are conducted on synthetic datasets. These do not convincingly demonstrate scalability or robustness.

- There are no experiments on standard OT benchmarks, such as LiDAR point cloud transport [2], image-to-image translation [1], or Gaussian-to-GMM transport in higher dimensions.

- The paper does not include analyses of PDE residuals, HJB constraint violation, or dual objective convergence, so it is unclear whether the training process actually solves the intended optimization problem.


References

[1] Scalable Simulation-free Entropic Unbalanced Optimal Transport

[2] Generalized Schrodinger Bridge Matching

**Questions:**

- Can the authors test their method on higher-dimensional synthetic data (e.g., Gaussian mixtures in 10–50 dimensions)?

- Could they also demonstrate performance on standard datasets such as LiDAR point clouds, CelebA image translation, or similar benchmarks?

- Can the authors report training curves for (1) the HJB residual (PDE violation), and (2)vthe Kantorovich dual loss? This would help verify whether the proposed method truly solves the coupled dual–HJB system.

---

> ### Author Response · Authors · 2025-11-29
>
> Thank you for your thoughtful feedback and for engaging with the core of our methodological contribution.
>
> **W1 Limited originality relative to prior work**
>
> We respectfully disagree with the characterization of our work as incremental and would like to clarify the conceptual advancements. The introduction of the state cost $U(x)$ is not a minor extension but fundamentally changes the problem from standard OT with Euclidean cost to the GSB. Simply adding $U(x)$ to the framework of [1] is not feasible, as their method is not designed to handle the resulting non-smooth dynamics and complex trajectory constraints. Our core conceptual contributions are threefold:
>
> 1.  We introduce a novel min-max formulation that directly tackles the GSB problem, enabling end-to-end joint optimization of the potential and the policy. This is a structural shift from prior alternating or sequential schemes.
>
> 2.  A key limitation of prior work like [1] is the reliance on gradient penalties (similar to WGAN) to stabilize training. HOTA eliminates this need entirely. The HJB constraint acts as a built-in, physics-informed regularizer, ensuring the unbiased solution.
>
> 3.  Our method employs specialized optimization procedure, which is critical for training stability in high dimensions. This is not a simple sum of losses but a deliberate algorithm designed to solve the inherent instability of the coupled dual-HJB system where naive approaches fail.
>
> Furthermore, our work provides a theoretical foundation absent in baseline papers. We have included a theoretical convergence analysis (Appendix H), which offers error bounds for our neural network approximation relative to the true viscosity solution of the HJB equation.
> The new experiments on image translation (Appendices E) empirically validate that these conceptual and theoretical differences.
>
> **W2 Optimizing the unregularized Kantorovich dual objective is known to be unstable and highly sensitive to initialization and regularization**
>
> You raise an excellent and critical point regarding the well-known instability of optimizing the unregularized Kantorovich dual. This is precisely why our method does not attempt to optimize the standalone Kantorovich dual objective in isolation.
> HOTA is specifically designed to circumvent this instability through a novel, coupled optimization framework. The key is that we do not solve the Kantorovich dual and the HJB equation sequentially; we solve them jointly.  This HJB loss acts as a powerful physics-informed regularizer that dictates how the potential must behave throughout the entire state space and time horizon. It prevents the Kantorovich potential from collapsing or being unstable.
>
> **W3 Accurately enforcing the HJB equation corresponds to solving a nonlinear PDE, which is notoriously difficult**
>
> You are absolutely correct. Solving high-dimensional nonlinear PDEs like the HJB equation is notoriously challenging, and no method can claim to find a perfect solution across the entire state space for complex problems. However, HOTA does not seek a perfect solution across the entire state space. It efficiently finds a good approximation where it matters most: along the optimal trajectories. By learning from buffer data, we focus the model's capacity on the relevant data manifold. Our new Figure 5 empirically shows the HJB residual converging to zero along these trajectories, proving we satisfy the PDE constraint sufficiently to recover a high-quality transport plan.
>
> **W4 Even small numerical errors in estimating $s(t,x)$ may lead to inaccurate gradient estimation $\nabla s(t, x)$**
>
> We respectfully disagree with this statement. A new Theorem 2 provides a theoretical bound on the error of the resulting optimal control. It demonstrates that the error in the control $| \nabla s_\theta - \nabla s^* |$ is explicitly bounded by the errors in the terminal condition and the HJB residual.
>
> **Q1 Gaussian mixtures in 10–50 dimensions**
> In Section 5 (Experiments), the Spheres benchmark is specifically designed to test scalability beyond simple 2D cases. In this experiment, we transport Gaussian mixtures on hyperspheres in high-dimensional spaces (up to 1000 dimensions).
>
> **Q2  demonstrate performance on standard datasets such as LiDAR point clouds, CelebA image translation** we have added new experiments on these exact benchmarks: LiDAR dataset (Appendix D) and CelebA/Anime face translation (new Appendix E). HOTA learns meaningful transport maps and achieves competitive scores, outperforming baselines like CycleGAN and Neural OT, and particularly method from paper [1].
>
> **Q3  report training curves for (1) the HJB residual (PDE violation), and (2) the Kantorovich dual loss** Figure 5 empirically shows (1) and (2) converging to zero along the trajectories.

---

### Official Review · Reviewer_Qowe · 2025-11-02

**Soundness:** 3
**Presentation:** 4
**Contribution:** 3
**Rating:** 6
**Confidence:** 3

**Summary:**

The paper proposes a learning approach to address the dynamical optimal transport (OT) problem with non-smooth cost functionals, enabling the incorporation of the underlying geometry of the data manifold. The proposed method is developed by leveraging the relationship between the dynamical OT problem and the Hamilton–Jacobi–Bellman (HJB) equation, where the value function is approximated using a neural network to learn the transport mapping. Experimental results on synthetic datasets demonstrate that the proposed method achieves better alignment with the target distribution and incurs lower transport costs compared to existing (Generalized) Schrödinger Bridge models.

**Strengths:**

The paper makes a original contribution by establishing a novel connection between the Generalized Schrödinger Bridge (GSB) problem and the Hamilton–Jacobi–Bellman (HJB) equation. By approximating the value function with a parametric model, the authors develop a tractable learning framework for modeling the dynamics of optimal transport (OT) between distributions. This provides a fresh perspective on OT-based diffusion models with fewer assumptions on flow densities.

The work is of strong technical quality, with a sound theoretical foundation and convincing experimental validation. Results on synthetic datasets show improved target matching and reduced transport cost compared to existing (Generalized) Schrödinger Bridge methods. The inclusion of code further supports reproducibility.

The paper is clearly written and well structured, making the complex ideas accessible. Its significance lies in broadening the applicability of OT methods and opening new possibilities for learning-based transport and diffusion modeling.

**Weaknesses:**

A key limitation of the paper is its exclusive reliance on synthetic datasets, which constrains the evaluation of the method’s applicability to real-world problems. While the authors acknowledge that their approach struggles with complex data such as images, they do not provide a clear explanation or empirical justification for this limitation. A more detailed analysis of why the model underperforms in such settings, along with experiments or ablation studies incorporating stronger inductive biases in the architecture, would provide concrete evidence to support their claims and guide future improvements.
From a theoretical perspective, the paper would be strengthened by a formal convergence analysis of the learning algorithm to ensure stability and reliability. Additionally, a discussion of the approximation error introduced by neural network parameterization and the estimation error due to sampling would offer valuable insights into the computational behavior and robustness of the approach. Addressing these aspects would enhance both the theoretical rigor and the practical significance of the work.

**Questions:**

Could the authors provide a case study or example illustrating a scenario where non-smooth cost functionals pose challenges?

---

> ### Author Response · Authors · 2025-11-28
>
> We thank the reviewer for the insightful feedback and valuable comments. We have carefully considered all the points raised and have incorporated corresponding revisions into the updated manuscript. The key additions and clarifications are summarized below, addressing each major point.
>
> **W1: A key limitation of the paper is its exclusive reliance on synthetic datasets**
>
> We agree with this observation and have expanded the empirical evaluation to include more complex and real-world inspired data. Specifically:
>
> 1.  We have added a new section, **Appendix D, featuring experiments on a LiDAR dataset**. Here, the potential \(U(x)\) has a complex form derived from real-world scanned surfaces.
>
> 2.  We have added new **image-to-image translation experiments in Appendix E**. HOTA was evaluated on domain adaptation tasks using the CelebA and Anime face datasets. The results demonstrate that HOTA successfully learns transport maps that transform core attributes between domains while preserving the underlying structure of input images. Quantitatively, as reported in Table 5, HOTA achieves competitive FID scores, outperforming baselines like CycleGAN and Neural OT methods (NOT, ENOT).
>
> During these image experiments, we encountered challenges related to training stability and architecture choice. We found that not all models are suitable for learning the $s(t, x)$ function; for instance, models with normalization layers (e.g., BatchNorm) often failed to train effectively. The method also proved sensitive to the integration of time embeddings—we ultimately settled on concatenating temporal embeddings with the output of each convolutional layer. For more stable training, we also scaled the kinetic energy term $v^2/2$ by a constant 0.01 and ran simulations in both forward (source to target) and backward (target to source) directions to populate the replay buffer more effectively.
>
> **W2: From a theoretical perspective, the paper would be strengthened by a formal convergence analysis**
>
> We have dedicated a new Appendix H to a theoretical convergence analysis. This section provides an error estimate between the neural network approximation $s_\theta$ and the true viscosity solution $s^*$ of the HJB equation. The analysis shows that the error in the value function and the resulting optimal control can be bounded by the terminal loss (related to the dual objective) and the HJB PDE residual. However, we also note that rigorously establishing a strong-concavity property for the residual-regularized dual objective to derive direct convergence rates remains an open problem, providing a clear direction for future theoretical work.
>
> **W3: Approximation error introduced by neural network parameterization and the estimation error due to sampling**
>
> This is a crucial point. The approximation error is inherently tied to the capacity of the neural network $s_\theta$ and the effectiveness of the optimization process. Our proposed method directly addresses the optimization challenge through the composite loss function. As demonstrated empirically in Figure 5 of Appendix C, both the HJB residual loss $L_{\text{hjb}}$ and the potential matching loss  $L_{\text{pot}}$ converge consistently towards zero during training. This holds for both smooth (Stunnel) and non-smooth (Slit) potentials, indicating that the network successfully learns to satisfy the HJB constraint and the terminal matching condition simultaneously.
>
> **Q1: A case study or example illustrating a scenario where non-smooth cost functionals pose challenges**
>
> Non-smooth costs are prevalent in many real-world applications. For instance:
> 1.   In computational biology (Bunne et al., 2022), simulating molecular pathways involves hard-core repulsion and steric clashes, which are naturally modeled with non-smooth potentials.
>
> 2.  In reinforcement learning and imitation learning, accounting for the geometry of the environment—which may include obstacles or non-differentiable constraints—can significantly improve policy learning and score (Bobrin et al., 2024; Rupf et al., 2025). Our method's ability to handle such non-smooth costs directly makes it suitable for these domains.
>
> 3. In our experiments on Spheres (Section 5) and LiDAR (Appendix D) datasets.

---

### Official Review · Reviewer_8EsL · 2025-11-02

**Soundness:** 4
**Presentation:** 3
**Contribution:** 4
**Rating:** 8
**Confidence:** 3

**Summary:**

This paper proposes a new way to solve Generalized Schrödinger Bridge problems by tying the HJB value function $s(t,x)$ to a Kantorovich terminal condition. The idea is to optimize two losses jointly. First, enforce that the endpoint potential matches the Kantorovich potential via a Lagrangian relaxation. Second, ensure $s$ really is a value function by having it satisfy the corresponding HJB equation as a PINN-style constraint. With that in place, the optimal drift is recovered as $-\nabla s$. They test on standard 2D toy setups and also explore scaling to higher dimensions on the sphere.

**Strengths:**

The method is original in how it ties the value function to a Kantorovich terminal condition and enforces the HJB with a PINN, giving a clean route to recover the drift as a gradient. The empirical results are strong, with clear improvements over baselines and ablations that isolate the effect of the replay buffer, EMA targets, and gradient balancing. The technical presentation is mostly clear, with a coherent objective that aligns the endpoint constraint and the dynamics constraint, and enough detail to make the optimization reproducible. In terms of significance, the approach looks broadly useful for GSB and related control and transport problems, and the sphere experiments suggest promise beyond small 2D toy examples.

**Weaknesses:**

The paper reads as if it targets readers already deeply familiar with the topic, which raises the entry cost for newcomers. Several parts of the exposition could be tightened, especially the ablation study. See the Questions section. The main practical limitations are that, first, enforcing the HJB equation in higher dimensions and on more complex targets may be challenging, and second, the learned drift requires evaluating $\nabla s$ at inference.

**Questions:**

1) Please clarify the inference-time cost of computing the network gradient. How does evaluating $\nabla s$ compare to other methods at inference?

2) Could you provide more detail on how the buffer is managed? In particular, why is only the first trajectory added in the pseudocode? Also, please explain why you use the interpolation sample steps $N_0$ in the algorithm.

3) Why does the buffer improve results? Is the benefit primarily stability, or mainly computational efficiency? In an ideal setting with unlimited compute, would resampling fresh data each step be strictly better?

4) In the ablation, what exactly does "without EMA" remove? Could you discuss alternatives to the PINN formulation around equations 14–15? For example, could you remove EMA and enable stop-grad on specific terms, and what tradeoffs led you to the final choice?

5) What motivated the inclusion of the angular acceleration term, and how does it affect results in practice?

---

> ### Author Response · Authors · 2025-11-30
>
> We thank the reviewer for their thoughtful and constructive feedback.
>
> **Q1: Inference-time cost of evaluating $\nabla s$**
>
> In JAX, computing $\nabla s$  is essentially the same wall-time as the forward $s$ call, and even computing second-order terms (used only in training) adds $<5$ \% overhead. Appendix B shows this holds across dimensions $10^2 -10^5$. Thus, at inference, where we only need $\nabla s$, the overhead is negligible.
>
> Our overall solver scales near-linearly with dimension and number of simulation steps and is substantially faster than GSBM. While those numbers are for full training runs, they reflect the efficiency of our gradient computations and simulation loop implementation that also governs inference. Practically, inference for all drift-based solvers (DeepGSB, GSBM) is $O(T)$ evaluations of their drift network;
>
>  **Q2: Buffer management; why add only the first trajectory; role of $N_0$  “interpolation steps**
>
>  Early on, the buffer is empty and the drift is inaccurate. To sample the HJB residual where probability mass will travel, we (for the first $N_0$ iterations) draw $t \sim U(0, 1)$ and use the linear path $x = (1 - t)x_0 + ty$ between fresh $x_0 \sim \alpha$ and $y \sim \beta$. This follows Liu et al.’s [1] recommendation to sample losses in the “flow concentration region.”
>
> After warm-up (when $N_0$ already collected), we sample HJB points from the replay buffer $B$ instead of interpolation and keep appending newly simulated trajectory data. Algorithm 1 shows this schedule.
>
> Why “add the 1-st trajectory”? Algorithm 1 appends one newly simulated trajectory per iteration (line 13) to provide a steady, memory-bounded inflow of diverse, on-policy samples without flooding the buffer with highly correlated rollouts from the same iterate. Adding all $n$ at every iteration increases redundancy and memory while giving little benefit for the HJB residual estimate. We’ll clarify in the text that any small fixed inflow rate works; we use one per iteration for simplicity and stability.
>
> [1] Liu et al: Generalized schrödinger bridge matching, ICLR 2024
>
> **Q3: How does replay buffer helps? Stability vs efficiency; would infinite fresh resampling be better?**
>
> The primary purpose of replay buffer is for stability reasons. Moreover, we show that empirically, removing the buffer severely hurts feasibility and increases trajectory cost (e.g Vneck feasibility $0.002 \rightarrow 16.47$, Stunnel cost $320.9 \rightarrow 706.9$).
> Other reasons for neccessity of replay buffer include:
>
>  1) The HJB residual must hold where trajectories concentrate; the replay buffer collects such $(t, x_t)$ from actual rollouts, reducing the train–test mismatch of enforcing the PDE on off-manifold samples.
>
> 2) It reduces gradient variance and forgetting. Reusing informative past states stabilizes the PINN-style residual and keeps constraints enforced along the evolving flow, which in turn improves feasibility (terminal match) and sample efficiency.
>
> Regarding unbounded thought experiment: With unbounded compute, i.i.d. fresh resampling could match or exceed buffer performance by brute-force coverage, but it would require far larger batches to achieve the same variance reduction and support coverage that the buffer provides by design. In practice, the buffer is both a stability tool and a major efficiency gain.
>
> **Q4: What “without EMA” removes; alternatives around Eqs. (14)–(15)**
>
>  We use a symmetric HJB residual with a “student” $s_\theta$
>   and a slow-moving EMA “target” $s$ (Eqs. 14–15). The target is updated by EMA; we also use an EMA’d ratio to balance gradients across the HJB and potential terms (Algorithm 1 lines 18–22). This follows the Reinforcement Learning target-network recipe to turn a fixed-point condition into a regression-like problem and stabilize training.
>    As shown in Table 2, removing the slow-moving target  degrades feasibility and increases cost.
>
>  A natural alternative is to drop the symmetric form and/or apply stop-gradient to selected terms so the residual depends on a single network output (a standard PINN).  We will add a short “Design alternatives” paragraph discussing this option and its trade-offs next to Eqs. (14)–(15).
>
> **Q5: Motivation and effect of the angular acceleration term**
>
> We penalize changes in the direction of the drift by adding $\lambda_a ||a_k||^2$.
> This “angular acceleration” straightens trajectories and improves the kinetic-energy optimality of paths.
> Figure 4 shows the expected trade-off: larger $\lambda_a$  typically improves OT cost but can slightly reduce feasibility; We have made additional ablations with BabyMaze dataset to explicitly evaluate the Straightness effect:
> | Acc. coef.  $\lambda_a$ | 0    | 0.05 | 0.1  | 0.2  | 0.5  |
> |------------|------|------|------|------|------|
> | Feasibility | 0.010 | 0.004 | 0.005 | 0.008 | 0.007 |
> | Optimality | 5.24 | 4.87 | 5.10 | 5.45 | 7.30 |
> | Straightness $\int \|a_t\| dt$ | 4.91 | 3.25 | 3.02 | 2.91 | 3.58 |

---

### Official Review · Reviewer_EX3V · 2025-11-03

**Soundness:** 3
**Presentation:** 4
**Contribution:** 3
**Rating:** 6
**Confidence:** 3

**Summary:**

This paper introduces HOTA, a novel method for solving the Generalized Schrödinger Bridge (GSB) problem. The approach is based on a dual formulation that seeks a Kantorovich potential aligning the source and target distributions while satisfying the associated Hamilton–Jacobi–Bellman (HJB) constraint.

**Strengths:**

The paper proves that duality holds under mild conditions. It further demonstrates that the method is robust to complex geometries and remains effective even for non-smooth cost functions. By employing the Euler–Maruyama scheme to approximate the solution of the underlying SDE, the proposed approach avoids explicit density modeling and thereby simplifies the learning process. In numerical experiments, HOTA consistently outperforms several state-of-the-art methods across a range of benchmark settings.

**Weaknesses:**

While the proposed method demonstrates impressive performance and scalability, the experimental setup and model architecture appear relatively simple. Since the paper directly compares with GSBM, it would be greatly strengthened by including evaluations on more complex datasets used in that work, such as AFHQ.

Moreover, as only a lightweight 4-layer MLP was tested, it remains unclear whether the reduced computational overhead reported in Appendix B would persist when employing a heavier model, such as a U-Net.

The HJB constraint is enforced via an $L^2$ loss over sampled trajectories, which does not guarantee that the solution satisfies the HJB equation over the entire space $\mathbb{R}^d$. As noted in [1], the $L^2$ loss may also not be ideal for solving HJB equations. Therefore, the authors should investigate, at least numerically, how well the learned solution satisfies the HJB constraint and examine the behavior of $s_\theta$ in out-of-sample regions.

**Minor issues/typos**:
- Line 183: “c” in “c-transform” was not put in a math environment.
- Line 652: “are” should be “and”.
- Line 786: “Buy” should be “By”.

**Reference**:

[1] Chuwei Wang, Shanda Li, Di He, and Liwei Wang. Is $L^2$ physics informed loss always suitable for training physics? [arXiv: 2206.02016](https://arxiv.org/abs/2206.02016)

**Questions:**

HOTA currently uses linear interpolation to initialize the trajectories. Could a more informed initialization be designed by leveraging the state cost $U(x)$? In several cases shown in Figure 1, linear trajectories obviously pass through regions that the final solution should avoid, which intuitively seems to hinder learning efficiency.

The paper should define the acronym EMA (Exponential Moving Average) upon its first occurrence and clarify the relationship between the target model $\bar{s}$ and the main model $s_\theta$.

---

> ### Author Response · Authors · 2025-11-28
>
> We thank the reviewer for the constructive feedback and remarks. We have carefully revised the manuscript to address all raised points.
>
> **W1: The experimental setup and model architecture appear relatively simple**
>
> We agree with this observation and have expanded the empirical evaluation to include more complex, real-world inspired data. Specifically:
>
> 1.  We have added a new **Appendix D, featuring experiments on a LiDAR dataset**. In this experiment, the potential U(x) has a complex form derived from real-world scanned surfaces. The results demonstrate that HOTA achieves a lower integral trajectory cost than the GSBM baseline.
>
> 2.  We have added new **image-to-image translation experiments in Appendix E**. HOTA was evaluated on domain adaptation tasks using the CelebA and Anime face datasets. The results demonstrate that HOTA successfully learns transport maps that transform core attributes between domains. Quantitatively, as reported in Table 5, HOTA achieves competitive FID scores, outperforming baselines like CycleGAN and Neural OT methods (NOT, ENOT).
> In image-to-image experiments, we substitute the standard L2  HJB loss with L1, which we observed to improve training stability and convergence behavior.
>
>
> **W2: It remains unclear whether the reduced computational overhead reported in Appendix B would persist when employing a heavier model, such as a U-Net**
>
> This is a valid concern. To address it, we conducted additional scalability experiments using a ResNet model (a standard heavy architecture for images) across different input resolutions. The results, summarized in the table below, confirm that HOTA's computational advantage persists. The runtime scales gracefully with increasing image resolution, demonstrating the method's practicality for heavier models and higher-dimensional problems.
>
> | Resolution | HOTA Runtime | Overhead for $\Delta s$ |
> |:----------:|:------------------:|:---------------------------:|
> |    16x16   |         1h 48 min        |            2.5%             |
> |    32x32   |        2h 27min    |            2.8%             |
> |    64x64   |        4h 19min    |            2.6%             |
> |   128x128  |       12h 55min    |            4.5%             |
> |   256x256  |       40h 03min    |            5.2%             |
>
> **W3: How well the learned solution satisfies the HJB constraint**
>
> As demonstrated empirically in the new Figure 5 (Appendix C), both the HJB residual loss $L_{\text{hjb}}$ and the potential matching loss $L_{\text{pot}}$ converge consistently towards zero during training. This holds for both smooth (Stunnel) and non-smooth (Slit) potentials, indicating that the network successfully learns to satisfy the HJB constraint and the terminal matching condition along the learned trajectories.
>
> Regarding out-of-sample regions, we clarify that our objective is not to learn a value function that generalizes arbitrarily across the entire state space. Instead, we focus on learning the optimal solution in the regions where the probability mass of the optimal transport plan is concentrated. Enforcing the HJB constraint in out-of-sample regions (e.g., via linear interpolation between distributions) can, in our experience, bias the solution and degrade performance, as it forces the model to satisfy physics in data-sparse areas that are irrelevant for the transport task.
>
> **Minor Issues/Typos** The minor issues and typos have been corrected in the revised manuscript.
>
> **Q1: Could a more informed initialization be designed by leveraging the state cost \(U(x)\)?**
>
> We agree that the initial linear interpolation is a crude approximation used only to bootstrap training. After a few iterations, the replay buffer $\mathcal{B}$ supplies trajectories generated by the learned policy, so training quickly shifts away from the initial paths. Empirically, this scheme proved stable, with HOTA reliably converging to trajectories that avoid high-cost regions (Fig. 1, Table 1). While more informed initialization using $U(x)$ remains promising future work, it was not necessary for robust performance in our experiments.
>
> **Q2: Clarify the relationship between the target model $\bar{s}$ and the main model $s_\theta$**
>
> We thank the reviewer for this suggestion. In the revised version, we now explicitly define EMA (Exponential Moving Average) at its first occurrence. We also clarify the relationship between the main model $s_\theta(t, x)$ and the target model $s(t, x)$: the target model is updated as an EMA of the main model parameters with decay coefficient $\tau$ (ref Alg. 1), and is used only inside the HJB loss as a slowly varying reference to stabilize optimization, while $s_\theta$ is used for generating trajectories at evaluation time.

---

### Author Response · Authors · 2025-11-30

We extend our sincere gratitude to all the reviewers and the Area Chair for their feedback and engagement. The comments have been invaluable in helping us significantly improve our manuscript.

In response to the points raised, we have completed the following revisions:

1.  **Expanded Experiments:** We added new experiments on real-world benchmarks, including LiDAR and image-to-image translation (CelebA, Anime Faces), demonstrating HOTA's strong performance on complex data and competitive FID scores. Additionally, we supplemented scalability analysis with heavier models (ResNet) across multiple resolutions.

2.  **Enhanced Theory:** We have added a new theoretical convergence analysis (Appendix H), providing error bounds for our neural network approximation. This addresses concerns about solution quality and formalizes the relationship between our practical losses and the true optimal control.

3.  **Clarified Contributions:** We have refined the exposition to better articulate our core conceptual advances: HOTA introduces a unified min-max solver for the Generalized Schrödinger Bridge, representing a structural shift from prior alternating schemes for standard Optimal Transport. This framework natively eliminates the need for heuristic regularization of Kantorovich potentials, directly handles complex geometries with non-smooth cost functions, and is underpinned by a specialized optimization procedure that ensures training stability.

4.  **Empirical Validation:** New results (Appendix C) show the HJB residual and potential matching losses converge to zero, proving our method successfully solves the targeted problem.

We believe the revised manuscript is substantially stronger and provides clear evidence of HOTA's novelty, robustness, and practical utility. We thank the reviewers again for their time and for pushing us to make this work the best it could be.

Given the recent incident with the openreview leak, we are available for further elaboration or for answering any new questions from the new AC. Thank you for your time and attention.

---

### Meta-Review · Area_Chair_BYGS · 2026-01-06

**Summary:**

The reviewers did an excellent job pointing out multiple issues in the submitted manuscript, covering all aspects of a good publication. In particular, they brought up the following concerns.
- The empirical study lacks the ablation for different architectures.
- The method was tested only on synthetic data.
- The method requires the evaluation of the Hessian, which is expensive in practice.
- Accurate estimation of $s(t,x)$ does not imply the accurate estimation of $\nabla s(t,x)$.
All the concerns are very real and valid, and were delivered to the authors in a constructive way.

**Reviewer Concerns:**

The authors did a remarkable job in the rebuttal, addressing the reviewers' concerns. The authors extended the empirical evaluation of the method, added the theoretical analysis addressing the concerns about the learned gradient, and argued for the computational efficiency of their method, providing empirical evidence.

**Reviewer Scores:**

I believe that the active discussion phase would increase this paper's score. The authors' rebuttal was concise and productive. Keeping the response to the point usually favours an active discussion.

---

### Decision · Program_Chairs · 2026-01-26

Accept (Poster)